# TPGS-based and S-thanatin functionalized nanorods for overcoming drug resistance in *Klebsiella pneumonia*

Xiaojuan Wang[1], Xiaoling Xu[2], Shaojun Zhang[1], Na Chen[1], Yunfeng Sun[2], Kuifen Ma[1], Dongsheng Hong[1], Lu Li[1], Yongzhong Du [2✉], Xiaoyang Lu [1✉] & Saiping Jiang [1✉]

Tigecycline is regarded as the last line of defense to combat multidrug-resistant *Klebsiella pneumoniae*. However, increasing utilization has led to rising drug resistance and treatment failure. Here, we design a D-alpha tocopheryl polyethylene glycol succinate-modified and S-thanatin peptide-functionalized nanorods based on calcium phosphate nanoparticles for tigecycline delivery and pneumonia therapy caused by tigecycline-resistant *Klebsiella pneumoniae*. After incubation with bacteria, the fabricated nanorods can enhance tigecycline accumulation in bacteria *via* the inhibitory effect on efflux pumps exerted by D-alpha tocopheryl polyethylene glycol succinate and the targeting capacity of S-thanatin to bacteria. The synergistic antibacterial capacity between S-thanatin and tigecycline further enhances the antibacterial activity of nanorods, thus overcoming the tigecycline resistance of *Klebsiella pneumoniae*. After intravenous injection, nanorods significantly reduces the counts of white blood cells and neutrophils, decreases bacterial colonies, and ameliorates neutrophil infiltration events, thereby largely increasing the survival rate of mice with pneumonia. These findings may provide a therapeutic strategy for infections caused by drug-resistant bacteria.

[1] Department of Clinical Pharmacy, The First Affiliated Hospital, School of Medicine, Zhejiang University, 79 Qingchun Road, Hangzhou 310003, China. [2] Institute of Pharmaceutics, College of Pharmaceutical Sciences, Zhejiang University, 866 Yu-Hang-Tang Road, Hangzhou 310058, China. ✉email: duyongzhong@zju.edu.cn; luxiaoyang@zju.edu.cn; j5145@zju.edu.cn

Antibiotics have been an epochal discovery for bacterial infection therapies in the past few decades. However, long-term and excessive use of antibiotics has caused the spread of antibiotic resistance[1]. Subsequently, the emergence of superbugs has been growing to become a dominant challenge in human health[2]. It is predicted that bacterial infections will lead to 10 million deaths each year by 2050, exceeding those presently caused by cancer[3]. Bacteria resist the effects of antibiotics mainly through the following molecular mechanisms: modification of the target site, destruction of the antibiotic, antibiotic efflux via efflux transporters and reduced antibiotic influx through decreasing membrane permeability[4]. Since its initial discovery in the 1980s, many efflux pumps have been characterized in pathogens such as *Staphylococcus aureus, Enterococcus faecium, Acinetobacter baumanii, Pseudomonas aeruginosa*, and *Klebsiella pneumoniae*[5]. Overexpression of efflux pumps can lead to clinically relevant levels of resistance to antibiotic in Gram-negative bacteria[6].

Currently, *Klebsiella pneumonia* (KPN) is regarded as one of the most serious nosocomial pathogens threatening public health[7]. As a gram-negative bacterium, KPN can induce multiple infections, such as pneumonia, liver abscesses, urinary tract infections, meningitis, and bacteremia in hospitalized patients with insufficient immune systems[8]. In recent years, KPN strains with high virulence and mucus phenotypes have gradually become an important pathogen clinically that can cause serious infections in healthy people and significantly threaten their health[9,10]. Moreover, KPN is prone to generating multidrug resistance[11]. Carbapenem antibiotics used to be the first-line drugs for infection caused by multidrug-resistant KPN (MDR-KPN). However, with the wide use of carbapenem antibiotics in clinical therapy, carbapenem-resistant KPN (CRKP) has distinctly increased and become a serious public health issue[12]. Increasing drug resistance and high mortality have posed dominant challenges for clinical therapy. There are currently few alternative drugs available for treating CRKP infections, with tigecycline (TIG) being one of the few remaining antibiotics[13]. Therefore, TIG is generally considered to be one of the last defensive line against CRKP. However, a decreased sensitivity and growing resistance of CRKP to TIG were observed with increased clinical applications, significantly threatening last-line therapy[14]. The time- and cost-consuming process of new antibiotic development results in the much slower emergence of new antibacterial drugs than that of bacterial resistance[15].

Recently, nanodrug delivery systems (DDSs) have emerged as novel therapeutic means for deadly infections[16,17]. DDS can provide an increased drug retention time in blood, a reduced nonspecific distribution, and targeted delivery of drugs at the site of infection[18]. Combination therapy of nanomaterials and antibiotics might contribute to a better therapeutic index. Therefore, they are regarded as promising candidates for combating MDR bacteria[19]. Numerous studies have constructed DDSs based on Ag, Au, Cu, Fe, Ti, and mesoporous silica nanoparticles to treat infectious diseases[17], and their clinical use in vivo is hindered by safety. More efficient nanomaterial-based delivery strategies, such as pH-triggered, enzyme-sensitive, and bacterial toxin-triggered DDSs, could potentially allow the release of antibiotics in a spatiotemporally controlled fashion and have gain attention worldwide[20]. However, this process is complex and sophisticated, and the released antibiotics lack specificity for bacteria. Therefore, a safe DDS with targeting efficacy to bacteria might be an effective strategy for the treatment of infection caused by TRKP.

As of now, several mechanisms have been identified that are associated with tigecycline-resistant *Klebsiella pneumonia* (TRKP). Most commonly, non-specific active resistance-nodulation-cell division (RND) efflux pumps such as AcrAB-TolC are overproduced[21]. Tigecycline MICs of > 2 mg/L have been associated with significantly increased *ramA* levels[22]. It was reported that the majority of CRKP isolates were resistant to tigecycline due to increased expression of the efflux pump gene *acrB*[21]. A retrospective study in China displayed that the elevated expression of *acrB* and *ramA* was found in ~90% of the tigecycline-resistant isolates clinically[23].

D-alpha tocopheryl polyethylene glycol succinate (TPGS), a nonionic surfactant, is extensively applied in DDSs. Numerous studies have revealed that TPGS functions as an inhibitor and substrate of P-glycoprotein and dramatically decreases the efflux of drugs[24]. Therefore, TPGS could reverse the multidrug resistance of tumor cells. These features make TPGS a feasible alternative biomaterial in DDSs for tumor therapy, especially in tumors resistant to chemotherapy drugs[25]. Our previous studies have confirmed the inhibitory activity of TPGS on efflux pumps in bacteria[26,27]. Therefore, the application of TPGS in DDSs might be an effective means to enhance TIG accumulation in TRKP and achieve effective therapy for TRKP infections.

S-thanatin (Ts) peptide is a short antimicrobial peptide (AMP) that exhibits lipopolysaccharide (LPS) binding affinity[28], suggesting its potential as a therapeutic strategy for infection caused by gram-negative bacteria. The interaction between the Ts peptide and LPS could promote the intercalation of Ts into the cytoplastic membrane, subsequently resulting in a leaky cytoplastic membrane and the disintegration of bacterial respiration and energization[29]. Ts peptide can kill bacteria in a membrane-dependent manner and has been shown to exhibit active antibacterial activity towards numerous gram-negative bacteria, including MDR bacteria[30,31]. Ts peptide-functionalized levofloxacin-loaded liposomes showed targeted drug delivery to bacteria and exhibited an excellent therapeutic effect in the septic mouse model induced by MDR-KPN[32]. Calcium phosphate nanoparticles (Cap) have drawn wide attention as potential carriers due to their inherent superior properties, including high drug loading efficiency, good biocompatibility, and excellent biodegradability[33,34]. Ts-modified Cap DDSs might further enhance TIG accumulation in bacteria. Pneumonia caused by KPN and TRKP is one of the most common infectious diseases clinically.

In this work, we design a TPGS-based and Ts peptide-modified Cap nanodrug delivery system for TIG loading, and construct Ts-TPGS/Cap/TIG (TTCT) for pneumonia therapy. After incubation with bacteria, Ts peptide modification can achieve targeted TIG delivery into bacteria. TPGS exhibits inhibitory activity on efflux pumps and the small particle size of the nanodrug delivery system is beneficial for penetrating the cell walls. Therefore, TIG accumulation inside bacteria can be enhanced to increase the antibacterial activity of TTCT. In addition, the synergistic antibacterial capacity between Ts and TIG further enhances the antibacterial activity of TTCT (schematic depiction shown in Fig. 1). Consequently, overcoming drug resistance and effective therapy for TRKP-induced pneumonia can be realized.

## Results

**Preparation and characterization of the Ts-TPGS/Cap/TG nanodrug delivery system.** Ts-TPGS (TT) conjugate was synthesized via the esterification reaction between carboxyl groups of Ts peptide and hydroxyl groups of TPGS. (Boc)$_2$O was used to protect the amino group of Ts, and a synthetic illustration is shown in Fig. 2A. The $^1$H-NMR spectrum of the TT conjugate displayed the characteristic peaks of both Ts (~7.7 ppm)[35] and TPGS (~3.5 ppm), indicating that TPGS was successfully modified by Ts (Fig. 2B). Then, a TT conjugate was used as the stabilizer to prepare Ts-TPGS/Cap nanorods via a hydrothermal method. The crystal structure and the chemical composition of

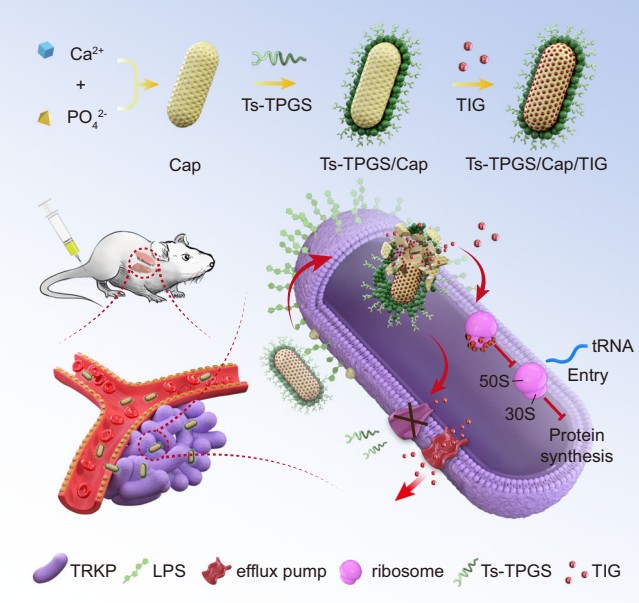

**Fig. 1 Schematic depiction of the fabrication of the nanorods and the mechanism of overcoming the tigecycline resistance of TRKP.** TIG tigecycline, Ts-TPGS/Cap/TIG tigecycline-loaded TPGS and S-thanatin functionalized calcium phosphate nanorods, TRKP tigecycline-resistant *Klebsiella pneumonia*, LPS lipopolysaccharide.

the nanorods were confirmed by comparing the X-ray diffraction (XRD) spectrum of nanorods with the typical crystal diffraction peaks of hydroxylapatite crystals (Fig. 3A). Afterwards, TIG was encapsulated into the Ts-TPGS/Cap and TPGS/Cap nanorods by the dialysis method. When the drug feeding ratio was 20%, the drug loading (DL %) was 12.47 ± 1.48 for TPGS/Cap/TIG (TCT) nanorods and 12.31 ± 0.94 for Ts-TPGS/Cap/TIG (TTCT). Correspondingly, the encapsulation efficiency (EE%) was calculated at 62.33 ± 6.92 for TCT and 61.53 ± 4.70 for TTCT.

The particle size, polydispersity index (PDI), and zeta potential of TPGS/Cap, Ts-TPGS/Cap, TCT, and TTCT were evaluated. The particle size of the formulations was maintained at ~25 nm (Table 1) and showed a narrow size distribution (Fig. 3B). No obvious changes were observed after Ts modification and TIG loading. All the formulations exhibited monomodal zeta potential distributions with a negative charge (Fig. 3C). However, the zeta potential of the nanorods decreased from −18.56 ± 1.66 mV to −14.99 ± 1.66 mV after Ts modification (Table 1), possibly because some hydroxyl groups of TPGS were replaced by Ts. Moreover, TIG loading further decreased the zeta potential, which was possibly due to the electrostatic interaction between the drug and nanorods. Transmission electron microscopy (TEM) images showed that all of the formulations had uniform and rod-like morphologies. (Fig. 3D). The stability of the formulations was explored as a function of time by dynamic light scattering (DLS). It was observed that the prepared nanorods were stable with small changes in particle size and PDI after storage for 20 days (Fig. 3E, F). Furthermore, the particle size and PDI of TCT and TTCT remained almost constant in 10% fetal bovine serum (FBS)-containing PBS within 24 h (as shown in Fig. 3G), indicating that TCT and TTCT would not disassemble in circulation.

The drug release behaviors of TCT and TTCT were evaluated by the dialysis method. As displayed in Fig. 3H, the nanorods released TIG at a rapid rate of ~25% in the first 0.5 h, then continued to release the drug for a sustained period of 72 h.

About 50% drug molecules got released at 8 h. The cumulative drug release was 86.89 ± 0.76 % and 87.80 ± 0.39 % for TCT and TTCT, respectively. Notably, the drug release profiles of TCT and TTCT were similar, indicating that Ts modification had no obvious effects on the drug release characteristics of the nanorods.

**Antibacterial activity of TTCT in vitro.** The in vitro antibacterial activity of TTCT was evaluated on *Klebsiella pneumonia* (KPN) and TIG-resistant KPN (TRKP) in our study. As shown in Fig. 4A, the minimal inhibitory concentrations (MICs) of TIG towards KPN and TRKP were 1 µg/mL and 4 µg/mL, respectively, indicating that TRKP was not susceptible to TIG. However, the Ts peptide displayed similar antibacterial activity against KPN and TRKP, with an MIC value of 16 µg/mL, which was consistent with previously reported studies[36]. In the case of KPN, TCT demonstrated similar antibacterial activity to TIG. Antibacterial capacity of TTCT was the best as evidenced by its lowest MIC value. It was possibly related to the combinational effect between TIG and Ts peptide. For TRKP, the MIC values of TIG and TCT were 4 µg/mL and 2 µg/mL, respectively, indicating that the TPGS-based nanodrug delivery system could ameliorate the TIG resistance of TRKP. Similar results were observed in previously reported studies and could be related to the activity of efflux pumps[26,27]. It is worth mentioning that the MIC value of TTCT was 1 µg/mL, which was equal to that of free TIG against KPN, suggesting that TTCT was effective in overcoming drug resistance.

We further investigated the effect of the formulations on the growth curves of KPN and TRKP. During the incubation period of 24 h with free TIG, TCT, and Ts-TPS/Cap (TIG concentration: 0.5 µg/mL; Ts-TPGS/Cap concentration: 5 µg/mL), a remarkable increase of the $OD_{600\ nm}$ values of KPN was observed, indicating that they could not prevent the growth of bacteria (Fig. 4B). However, visible growth of bacteria was completely inhibited by TTCT (TIG concentration: 0.5 µg/mL). In the case of TRKP, bacterial growth could only be inhibited by TTCT (TIG concentration: 1 µg/mL), with almost constant $OD_{600\ nm}$ values over 24 h of incubation. Growth of TRKP was not prevented by TIG (1 µg/mL, 2 µg/mL), TCT (TIG concentration: 1 µg/mL), and Ts-TPGS/Cap (10 µg/mL) (Fig. 4C). The result was in accordance with that shown in Fig. 4A. As a whole, TTCT demonstrated better antibacterial activity than TIG and TCT in both KPN and TRKP.

To further evaluate the antibacterial activity of TTCT, changes in bacterial morphology following the different treatments were observed by scanning electron microscopy (SEM). As displayed in Fig. 4D, bacteria without any treatment exhibited a typical clubbed shape and intact smooth surface. The bacterial morphology was obviously changed after treatment with various preparations. Ts-TPGS/Cap nanorods led to noticeable holes in their cell walls in both KPN and TRKP, which was possibly attributed to the antibacterial mechanisms of the Ts peptide. Additionally, the magnified SEM images of bacteria after incubation with TIG-containing preparations demonstrated significant morphological changes, such as broken cell membranes, debris, and loss of cellular integrity. The bacterial structures were massively destroyed by TTCT in both the KPN and TRKP groups, further confirming the superior antibacterial effect of TTCT. In conclusion, all of the results shown in Fig. 4 revealed that TTCT was a superior preparation with excellent antibacterial function against both KPN and TRKP when compared with TIG, TCT, and Ts-TPGS/Cap nanorods. Furthermore, the data suggested that TTCT was capable of overcoming the drug resistance of TRKP and achieved effective antibacterial capacity towards multidrug-resistant bacteria.

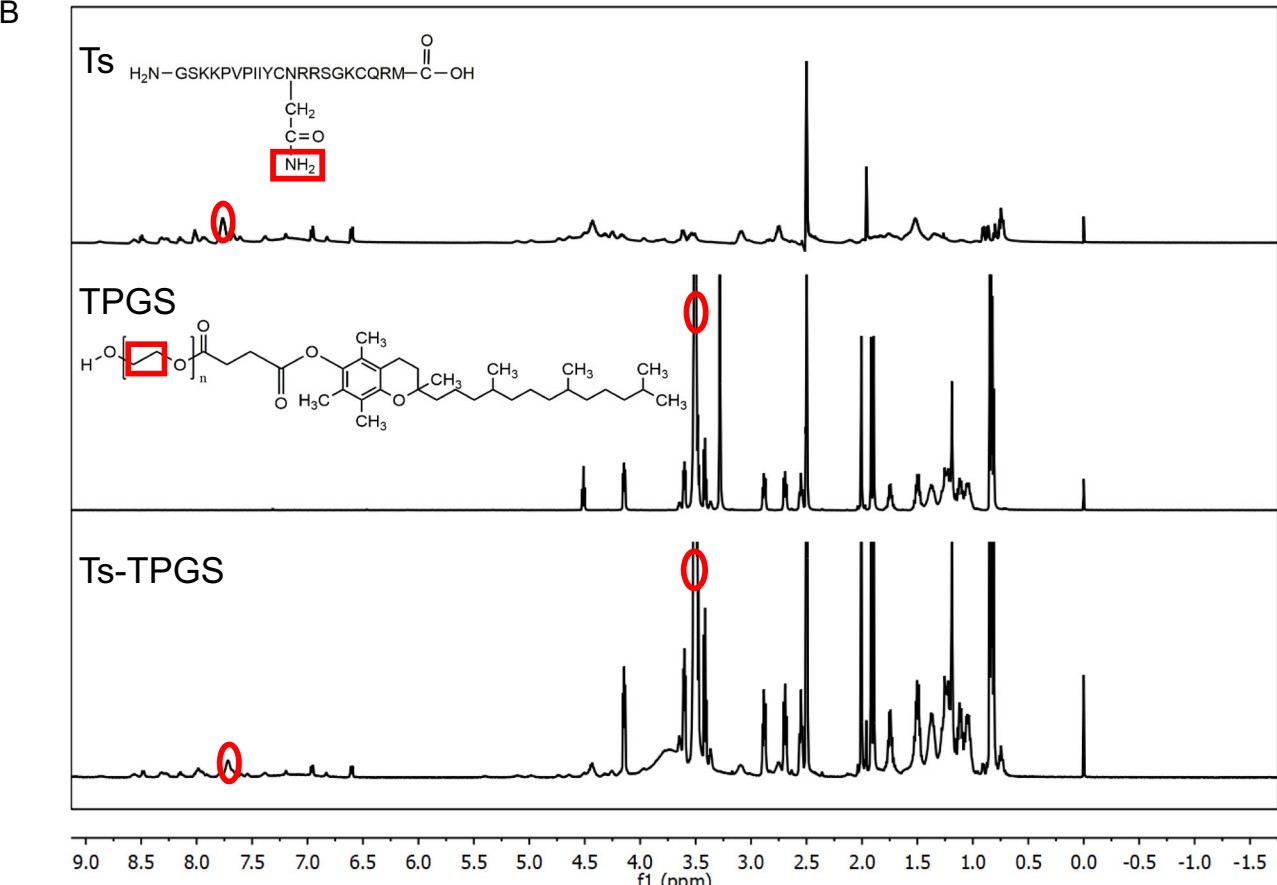

**Fig. 2 Synthesis and characterization of Ts-TPGS (TT). A** Synthetic scheme of the TT conjugate. **B** [1]H-NMR spectra of Ts peptide, TPGS, and TT from top to bottom. The red circles represented the characteristic peaks of the structure displayed in red squares, in which ~ 3.5 ppm represented the -O-CH2-CH2- of TPGS and ~ 7.7 ppm indicated the -CONH2 of Ts peptide. Ts S-thanatin peptide, TPGS tocopheryl polyethylene glycol succinate.

**Mechanism of overcoming drug resistance.** We speculated that the enhanced antibacterial activity of TTCT was attributed to the targeting capacity given by Ts, inhibitory effect on efflux pump endowed by TPGS, and synergy antibacterial activity between Ts and TIG. Below are the experiments we conducted to verify our hypotheses.

**In vitro targeting efficiency of Ts-TPGS/Cap nanorods.** It was reported that the β-hairpin structure of thanatin is stabilized by the disulfide bond between Cys11 and Cys18[37], which is considered integral for its activity[38]. An analog of thanatin with the two Cys residues replaced by Ala was found to be largely inactive[39], indicating that the disulfide bond is important for Ts

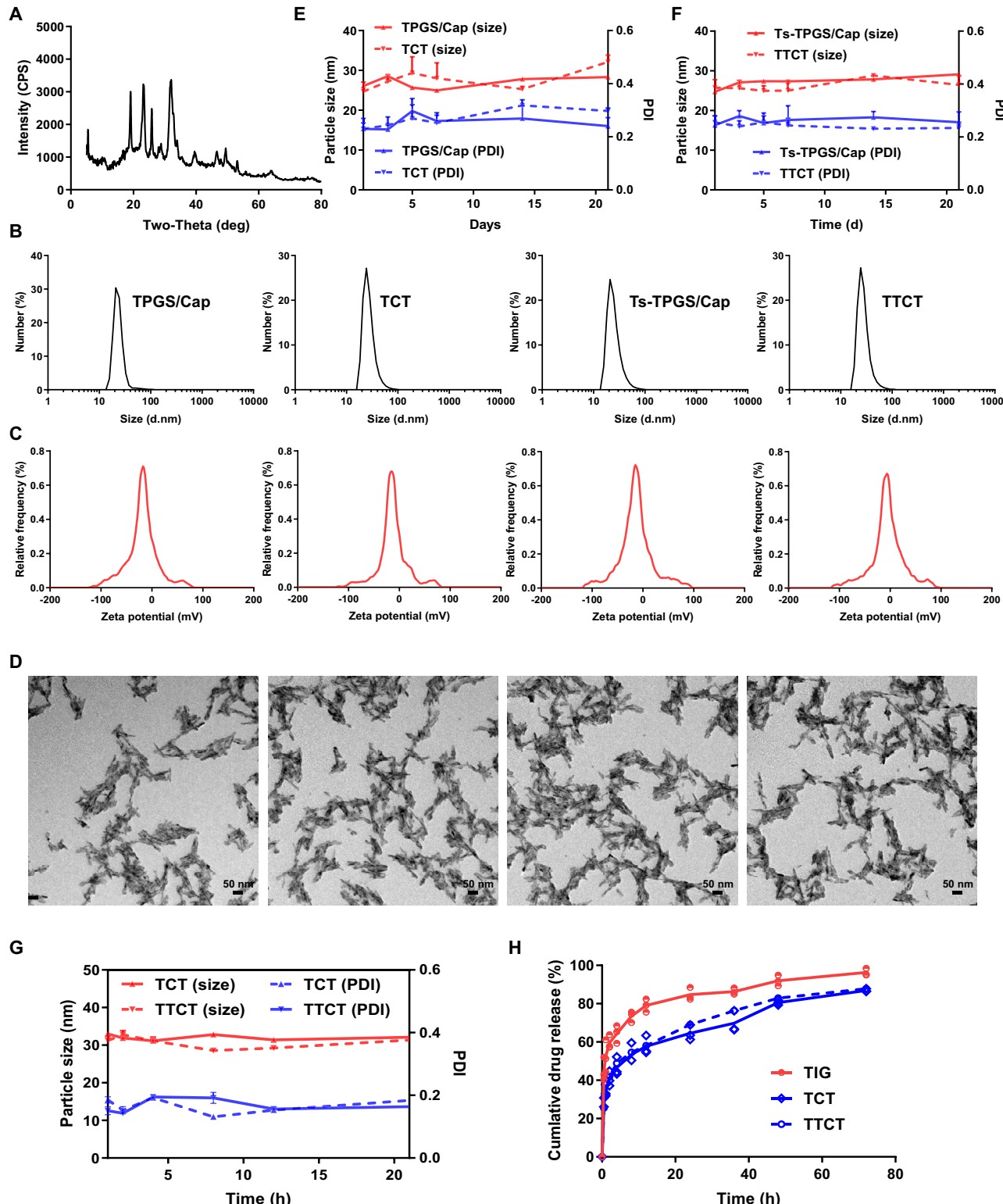

**Fig. 3 Preparation and characterization of TTCT. A** X-ray diffraction of TPGS/Cap nanorods. **B** Size distribution measured by DLS, **C** zeta potential distribution, and **D** representative TEM images of TPGS/Cap, Ts-TPGS/Cap, TCT, and TTCT (Scale bar = 50 nm). Particle size and PDI changes of **E** TPGS/Cap, Ts-TPGS/Cap, (**F**) TCT, and TTCT during 1 week of storage (n=3 independent experiments). **G** Changes in particle size and PDI of TCT and TTCT in 10% fetal bovine serum (FBS)-containing PBS (n = 3 independent experiments). **H** In vitro drug release profiles of TCT and TTCT in 10% fetal bovine serum (FBS)-containing PBS at pH 7.4 and 37 °C (n = 3 independent experiments). TIG free tigecycline, TPGS/Cap TPGS-functionalized calcium phosphate nanorods, TCT tigecycline loaded TPGS/Cap nanorods, TTCT tigecycline-loaded Ts-TPGS/Cap nanorods. Data are expressed as the mean ± SD, and the mean value is the average of three independent experiments. The experiments in **B**–**D** were repeated independently for three times with similar results. Source data are provided as a Source Data file.

activity. Considering this, an analog of Ts peptide with two Cys residues replaced by Ala (GSKKPVPIIYANRRSGKAQRM) was synthesized and served as the non-targeting peptide. The non-targeting peptide-conjugated TPGS/Cap (NT-TPGS/Cap) was prepared according to the preparation process of Ts-TPGS/Cap and used as control to evaluate the in vitro targeting efficiency of Ts-TPGS/Cap nanorods. As shown in Fig. 5A, both NT-TPGS/

Cap and Ts-TPGS/Cap nanorods displayed a gradual increase in the intra-bacteria fluorescent signal with a prolonged incubation time. Ts-TPGS/Cap nanorods displayed stronger bacterial internalization capacity than NT-TPGS/Cap nanorods in both KPN and TRKP at 2 and 6 h. The results were consistent with those obtained by flow cytometry (Fig. 5B, C). The fluorescent signals in KPN after incubation with Ts-TPGS/Cap nanorods were ~2-fold greater than those of NT-TPGS/Cap nanorods at both 2 h and 6 h (Fig. 5D). For TRKP, similar results were observed, suggesting that Ts-TPGS/Cap nanorods exhibited better bacterial internalization activity, which was possibly attributed to the targeting moiety Ts peptide.

**The inhibitory effect of TPGS on efflux pumps**. It has been reported that the accumulation of ethidium bromide (EB) is inversely correlated with the activity of efflux pumps[27]. Therefore, EB was exploited as a substrate of the efflux pump[40].

| Table 1 Characterization of TPGS/Cap, Ts-TPGS/Cap, TCT, and TTCT. | | | |
|---|---|---|---|
| Preparation | $d_n$ (nm) | PDI | $\zeta$ (mV) |
| TPGS/Cap | 27.1 ± 4.8 | 0.29 ± 0.03 | −18.56 ± 1.66 |
| Ts-TPGS/Cap | 25.9 ± 2.3 | 0.27 ± 0.01 | −14.99 ± 1.66 |
| TCT | 26.6 ± 3.9 | 0.24 ± 0.03 | −14.83 ± 2.28 |
| TTCT | 28.2 ± 3.3 | 0.29 ± 0.03 | −11.64 ± 1.97 |

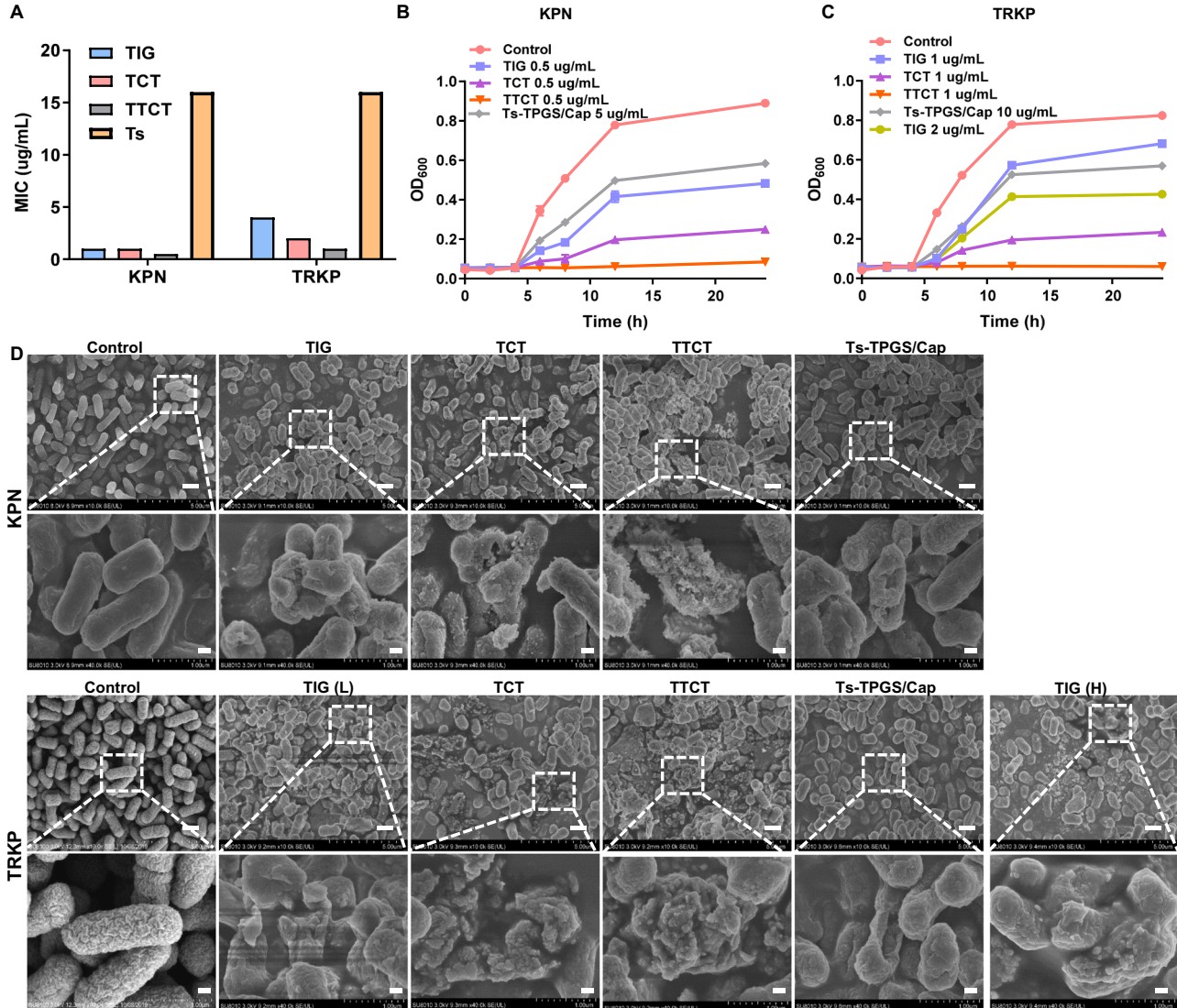

**Fig. 4 Evaluation of the antibacterial activity of TTCT towards KPN and TRKP in vitro. A** MIC susceptibility semiquantitative profiles of TTCT via the microplate broth dilution method ($n = 3$ independent experiments, and the results were similar). Growth curves of KPN (**B**) and TRKP (**C**) co-incubated with various preparations. **D** Representative SEM images of KPN and TRKP after treatment with TIG, TCT, TTCT, and Ts-TPGS/Cap nanorods. Scale bar on the original SEM images and the enlarged images were 10 μm and 2 μm, respectively. TIG free tigecycline, Ts-TPGS/Cap Ts-TPGS functionalized calcium phosphate nanorods, TCT tigecycline loaded TPGS/Cap nanorods, TTCT tigecycline-loaded Ts-TPGS/Cap nanorods, KPN *Klebsiella pneumonia*, TRKP tigecycline-resistant *Klebsiella pneumonia*. The experiments in **A**–**D** were repeated independently for three times with similar results. Source data are provided as a Source Data file.

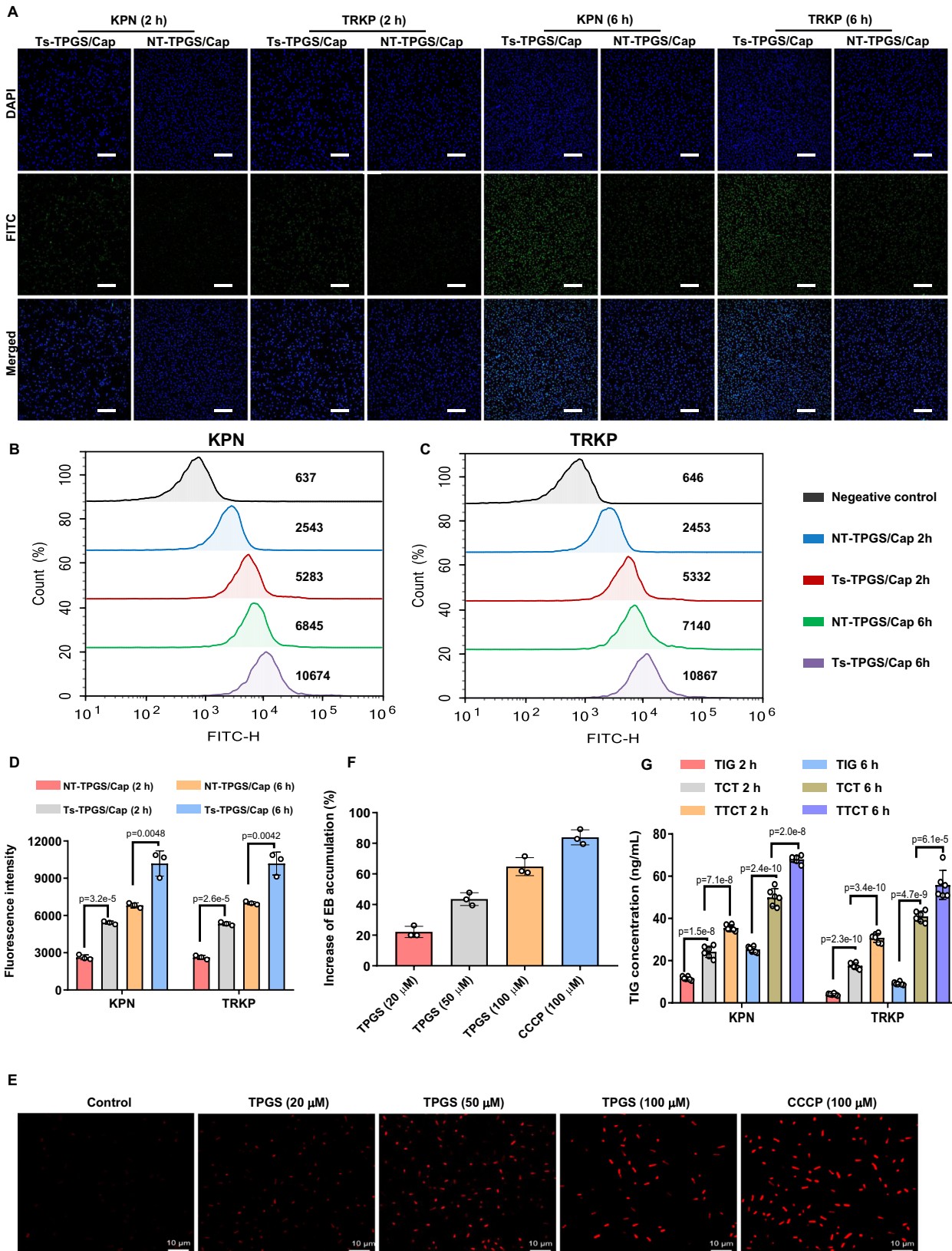

The inhibitory effect of TPGS on efflux pumps was investigated by comparing it to carbonyl cyanide m-chlorophenylhydrazone (CCCP), a well-known inhibitor of proton pump. As shown in Fig. 5E, EB accumulation in TRKP was negligible in the control group. After incubation with TPGS, the fluorescent signal of EB inside bacteria was significantly stronger than that in the control group. Quantitative analysis showed that the increased percentages of EB accumulation grew from $22.11 \pm 3.67\%$ to $64.75 \pm 5.91\%$ as the concentration of TPGS increased from 20 to 100 μM (Fig. 5F), indicating that TPGS treatment manifested an obvious increase in EB accumulation. Similar results were observed in previously reported studies[27].

**Fig. 5 Investigation of the mechanisms by which TTCT overcomes drug resistance.** Fluorescence images of KPN and TRKP after 2 h and 6 h of incubation with NT-TPGS/Cap and Ts-TPGS/Cap nanorods. The bacteria were labeled with DAPI (blue), and the green fluorescence signal indicated the nanorods. Scale bar = 20 μm. Fluorescence within KPN (**B**) and TRKP (**C**) detected by flow cytometry after incubation with NT-TPGS/Cap and Ts-TPGS/Cap. **D** Quantitative analysis of fluorescence inside KPN and TRKP after co-culture with nanorods. Data are expressed as the mean ± SD (n = 3 independent experiments). **E** Confocal fluorescent images of TRKP after exposure to EB for 4 h, following pretreatment with TPGS and CCCP. Scale bar = 10 μm. **F** Quantitative analysis of the increase in EB accumulation compared with the negative control groups (n = 3 independent experiments). **G** TIG concentration detected by HPLC-MS inside KPN and TRKP after incubation with TIG-containing preparations (n = 6 independent experiments). Ts-TPGS/Cap Ts-TPGS functionalized calcium phosphate nanorods, NT-TPGS/Cap non-targeting peptide-TPGS functionalized calcium phosphate nanorods, TIG free tigecycline, TCT tigecycline loaded TPGS/Cap nanorods, TTCT tigecycline-loaded Ts-TPGS/Cap nanorods, CCCP carbonyl cyanide m-chlorophenylhydrazone, EB ethidium bromide, KPN Klebsiella pneumonia, TRKP tigecycline-resistant Klebsiella pneumonia. Data are expressed as the mean ± SD. Unpaired two-tailed T-test was performed in **D**. One-way analysis of variance (ANOVA) with post hoc Tukey tests were performed in **G**. The experiments in A and E were repeated independently for three times with similar results. Source data are provided as a Source Data file.

In addition, tigecycline resistance in TRKP has been demonstrated to be associated with the overexpression of RND-type efflux pump genes (acrA and acrB) and ramA[41]. Real-time reverse transcription-PCR (RT-PCR) was used to determine the levels of acrA, acrB, and ramA. After bacteria harvesting in the mid-exponential phase, KPN and TRKP were exposed to TPGS, Ts, Ts-TPGS, and Ts-TPGS/Cap nanorods, respectively. As shown in supplementary Fig. 1, the expression levels of acrA, acrB, and ramA were up-regulated significantly compared with those in KPN, which confirmed that the AcrAB-TolC were involved in the TIG resistance of TRKP. Intervention of TPGS, Ts, Ts-TPGS, and Ts-TPGS/Cap obviously down-regulated the expression levels. It is worth mentioning that Ts treatment also down-regulated the expression levels of efflux pumps, and we speculated that the phenomenon was associated with the Ts activity on the outer membrane of the bacteria[29,42]. Ts-TPGS and Ts-TPGS/Cap nanorods displayed the highest reduction of efflux pump gene expression, possibly owing to the combinational effect of TPGS and Ts. Based on these results, we confirmed that TPGS had an inhibitory effect on the activity and expression levels of efflux pumps of TRKP. The results were consistent with those reported previously[26]. However, the underlying mechanisms remain poorly determined. As reported, surfactants can induce membrane fluidity alterations and ATPase inhibition[43]. Therefore, we conjectured that TPGS inhibited activity of efflux pump possibly via its surfactant characteristic.

**TIG concentration inside bacteria.** The targeting efficacy and the inhibitory effect on efflux pumps were beneficial for TIG delivery into bacteria. Therefore, the TIG concentration inside bacteria after KPN and TRKP incubation with free TIG, TCT, and TTCT was measured by high performance liguid chromatography- mass spectroscopy (HPLC-MS). As shown in Fig. 5G, the TIG concentration increased over time after the bacteria incubation with TIG, TCT, and TTCT. In the case of KPN, both TCT and TTCT led to a large increase in the TIG concentration than TIG group, and TTCT exhibited a higher concentration than TCT at both 2 h and 6 h. These results indicated that TIG delivery into bacteria was better after encapsulation in nanorods, and the effect was further enhanced by Ts peptide modification. Similar results were observed for TRKP. The TIG concentration in the TRKP group was only ~30% of that in the KPN group when the bacteria were incubated with TIG, which could be a reasonable explanation for the TIG resistance of TRKP. Combined with the results shown in Fig. 5E, we inferred that this was possibly because the efflux pumps would induce transport of TIG from inside to outside bacteria. This phenomenon was obviously ameliorated by the TPGS-based nanodrug delivery system. The Ts peptide modification further increased the TIG concentration inside bacteria. Overall, the higher TIG concentration of TTCT among these groups in TRKP was possibly due to the targeting

efficiency induced by Ts, inhibitory effect on efflux pump activity, and TIG encapsulation in nanorods.

**Synergy antibacterial activity between Ts and TIG.** The interaction between Ts and tigecycline was investigated by the microdilution checkerboard technique according to CLSI guidelines. The data exhibited that the fractional inhibitory concentration indices between Ts and tigecycline against KPN and TRKP were 0.75 and 0.375, respectively, indicating that Ts exerted synergistic effect with tigecycline against TRKP and additive effect against KPN (shown in supplementary Table 1).

Overall, the underlying mechanism of TTCT overcoming the drug resistance might include the following aspects: (1) The small particle size of the nanorods was beneficial for penetrating the cell walls; (2) TIG accumulation in bacteria was significantly enhanced via the targeting capacity of Ts to bacteria and the inhibitory effect on efflux pumps exerted by TPGS; (3) The synergistic antibacterial capacity between Ts and TIG further enhanced the antibacterial activity TTCT.

**Biodistribution in vivo.** Using tracheal injection of KPN or TRKP bacteria into the lungs of healthy mice, we created acute pneumonia mouse models and investigated the in vivo targeting efficiency of Ts-TPGS/Cap nanorods. The mice were intravenously injected with indocyanine green (ICG)-labeled nanorods at 5 h postoperation. Upon administration of Ts-TPGS/Cap nanorods, we evaluated the in vivo biodistribution of the nanorods 5 h and 24 h later. As shown in Fig. 6A, B, Ts-TPGS/Cap nanorods displayed similar bio-distribution characteristics between pneumonia mice infected by KPN and TRKP. The same phenomenon was also observed in mice treated with TPGS/Cap nanorods. Moreover, the fluorescence signals were much stronger in lungs treated with Ts-TPGS/Cap nanorods than in those treated with TPGS/Cap, implying that more Ts-TPGS/Cap nanorods accumulated in the infected lungs (Fig. 6A, B). In pneumonia mice caused by both KPN and TRKP, the quantitative signals of fluorescence (expressed by fluorescence intensity per gram of tissue, ID/g) of Ts-TPGS/Cap nanorods in the lung were ~1.5-fold and ~2.5-fold higher than those of TPGS/Cap at 5 h and 24 h, respectively (Fig. 6C, D). Notably, the Ts-TPGS/Cap nanorods were primarily distributed to the liver, lung, and kidney of the pneumonia mice at 24 h. The ID/g of the lungs were higher than those of the liver and kidneys.

Pneumonia may evoke disruption of pulmonary endothelial barrier integrity[44], and the vascular permeability may be significantly increased at the sites of bacterial infection[45]. It has been reported that the enhanced permeability and retention (EPR) effect might operate to deliver nanoparticles to bacterial-infected tissues[46]. We speculated that the nanorods we prepared might be distributed into the infected lung via EPR effect after intravenous administration. Then the nanorods could be

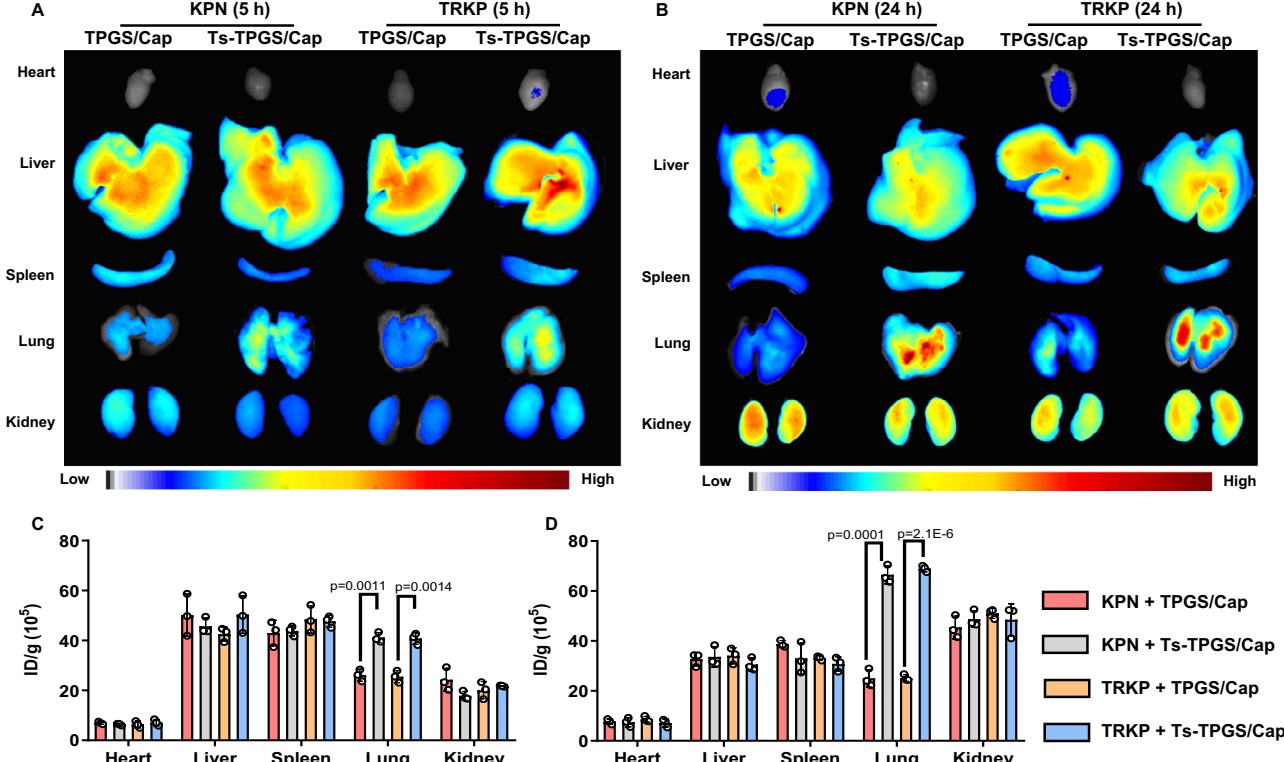

**Fig. 6 Evaluation of the biodistribution and targeting efficacy of Ts-TPGS/Cap in vivo.** Fluorescence images of major organs collected from KPN- and TRKP-infected mice with pneumonia (**A**) at 5 h (**B**) and 24 h postinjection of TPGS/Cap and Ts-TPGS/Cap nanorods. Semiquantitative analysis of the fluorescence intensity in major organs displayed in (**C**) Fig. 6A and (**D**) Fig. 6B (n = 3 mice). Ts-TPGS/Cap Ts-TPGS functionalized calcium phosphate nanorods, TPGS/Cap TPGS-functionalized calcium phosphate nanorods, KPN *Klebsiella pneumonia*, TRKP tigecycline-resistant *Klebsiella pneumonia*. Data are shown as the mean ± SD (n = 3 mice for each group). Unpaired two-tailed T-test was performed in **C**, **D**. Source data are provided as a Source Data file.

internalized by vascular endothelial cells (Data in supplementary Fig. 2 demonstrated that the nanorods had excellent cellular internalization capacity) and be transported across the blood vessel. The binding affinity between Ts and LPS promoted the accumulation of Ts-functionalized nanorods in the infected lung. Overall, an in vivo distribution study revealed that Ts-TPGS/Cap nanorods could realize targeting accumulation in the lungs of pneumonia mice, which was beneficial for achieving better anti-infective efficacy.

**Anti-infective efficacy of TTCT in vivo**. Acute KPN- and TRKP-infected pneumonia model was used to investigate the in vivo anti-infective activity of TTCT. After 5 h of modeling, the mice were intravenously administered TIG, TCT, TTCT, and Ts-TPGS/Cap nanorods. Then, the survival rate of mice receiving different treatments was recorded within 5 days. As shown in Fig. 7A, B, the pneumonia mice infected by TRKP survived 33.3% at 5 days, compared with the KPN-infected mice at 16.7%. This difference could be explained by the fact that after resistance is acquired, the bacteria may become less virulent[47,48]. KPN-infected mice receiving TIG and TCT treatment (TIG dose: 15 mg/kg) survived 66.7% and 83.3%, respectively, indicating that encapsulating TIG in nanorods boosted its antibacterial activity. Moreover, Ts-TPGS/Cap nanorod treatment also enhanced the survival rate of KPN-infected mice.

In contrast, the TRKP pneumonia mice receiving therapy of TIG (15 mg/kg) demonstrated a lower survival rate than the KPN-infected mice, possibly attributed to the nonsusceptibility of TRKP to TIG. 66.7% of TRKP-infected mice were alive after 45 mg/kg TIG treatment, but the rate was still lower than that of

mice given TCT (TIG dose: 15 mg/kg). This phenomenon suggested that TPGS-based nanodrug delivery systems could overcome bacterial resistance and achieve effective therapeutic effects against infection induced by drug-resistant bacteria. Combined with the data shown in Fig. 5E, F, the results could be explained by the finding that TPGS was capable of inhibiting the activity of efflux pumps. Additionally, the TTCT treatment group had the highest survival rate of 100% in pneumonia mice caused by both KPN and TRKP. The superior therapeutic effect compared with other groups might be attributed to the targeting distribution, inhibitory activity of efflux pumps, and Ts antibacterial peptide.

Inflammatory markers (white blood cell count, neutrophil count, and level of C reactive protein) are important indicators of inflammation that can be used to evaluate infection. Therefore, the survived mice receiving therapy of various preparations were sacrificed after 48 h for collection of blood samples and bronchoalveolar lavage fluid (BALF). First, we evaluated the levels of white blood cells (WBCs) and neutrophils in blood samples. The results showed that bacterial infection caused an obvious increase of WBCs and neutrophils (Fig. 7C–D). A significant reduction in the levels of WBCs and neutrophils was observed when the KPN- and TRKP-infected mice were treated with TIG, TCT, and TTCT, indicating that the infection was apparently alleviated. Moreover, a greater reduction in the levels of WBCs and neutrophils was observed in the TTCT group in both KPN- and TRKP-infected mice, demonstrating that TTCT had a superior therapeutic effect than the other treatments. It should be mentioned that Ts-TPGS/Cap nanorods alone had positive anti-infective activity towards KPN- and TRKP-infected mice. The reduction in WBC and neutrophil levels in the

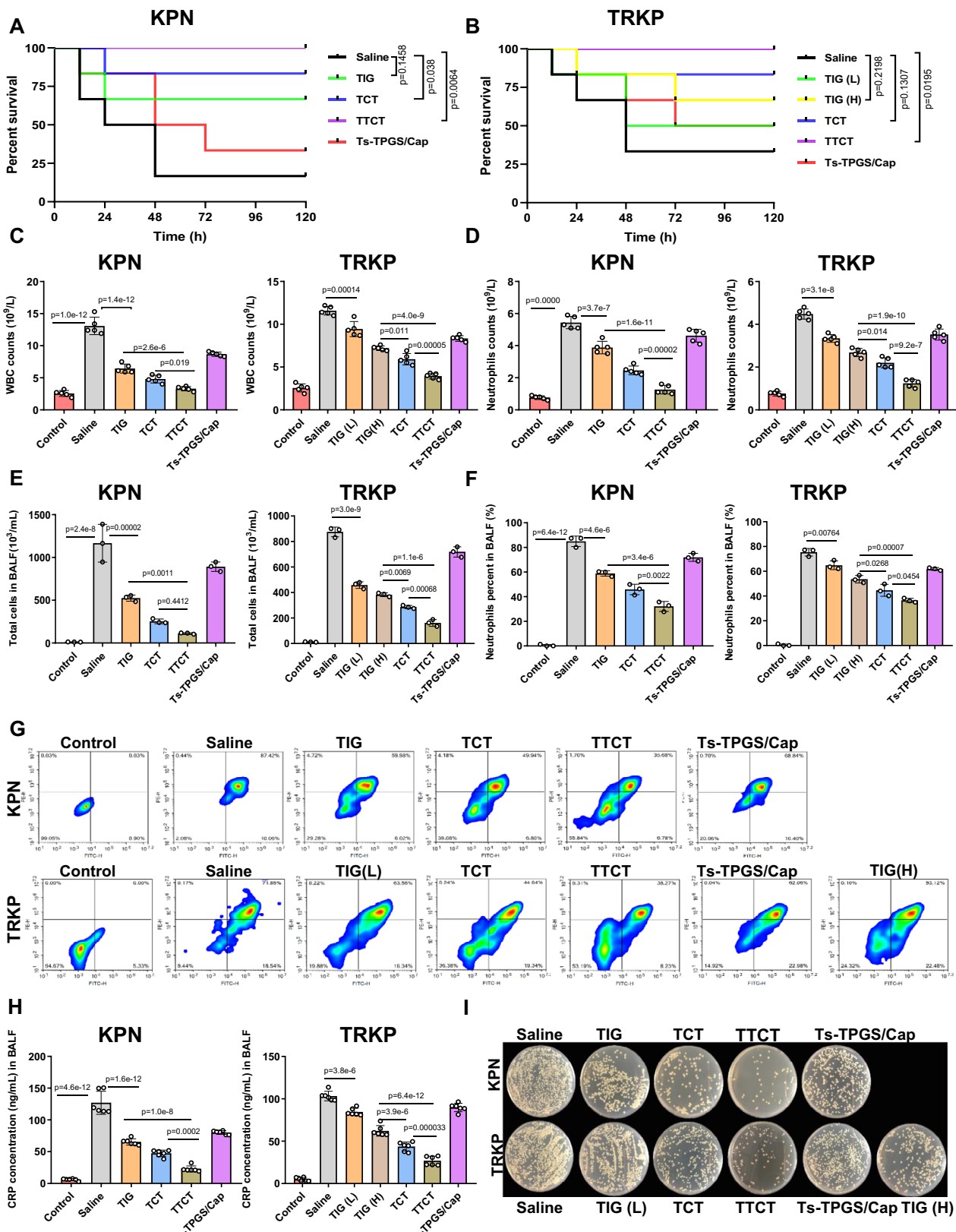

Ts-TPGS/Cap treatment group was even equal to that of the TIG (L) group (TIG dose: 15 mg/kg) in TRKP-infected mice. A possible reason for the phenomenon was the excellent anti-bacterial activity of the Ts peptide towards both drug-sensitive and drug-resistant bacteria. The results were consistent with those displayed in Fig. 7A, B.

Then, we investigated the levels of total cells, neutrophil percentages, and C reactive protein (CRP) in BALF to further evaluate the anti-infective efficacy of TTCT in vivo. As shown in Fig. 7E, the total cells in BALF excessively increased in KPN- and TRKP-infected mice. When the infected mice received TIG-containing preparations and Ts-TPGS/Cap nanorods, the total

**Fig. 7 Investigation of the anti-infective efficacy of TTCT in vivo. A** Survival rates of acute KPN-infected and (**B**) TRKP-infected mice with pneumonia treated with TIG, TCT, TTCT, or Ts-TPGS/Cap nanorods for 5 days ($n = 6$ mice per group). Mice with pneumonia treated with saline were used as a negative control. **C** WBC and **D** neutrophil counts in blood samples from mice with pneumonia receiving various preparations ($n = 5$ mice for each group). **E** Total cell counts and **F** neutrophil percentages in BALF obtained from KPN- and TRKP-infected mice with pneumonia administered different preparations ($n = 3$ mice for each group). **G** Flow cytometry results of anti-Ly-6G/Ly-6C and anti-CD11b double-stained cells obtained from the BALF of mice with pneumonia. Double-positive cells represented neutrophils in BALF. **H** Levels of CRP in the BALF of mice with pneumonia after different treatments ($n = 6$ mice for each group). **I** Representative KPN and TRKP bacterial colonies formed on LB agar plates from the BALF of mice with pneumonia receiving various treatments. TIG free tigecycline, TIG(L) low dose of tigecycline (15 mg/kg), TIG (H) high dose of tigecycline (45 mg/kg), TCT tigecycline loaded TPGS/Cap nanorods, TTCT tigecycline-loaded Ts-TPGS/Cap nanorods, KPN *Klebsiella pneumonia*, TRKP tigecycline-resistant *Klebsiella pneumonia*. Data are presented as the mean ± SD. Log-rank (Mantel-Cox) test was performed in **A** and **B**. One-way analysis of variance (ANOVA) with post hoc Tukey tests were performed in **C**, **D**, **E**, **F**, and **H**. Source data are provided as a Source Data file.

cells exhibited a significant reduction, and TTCT displayed the highest reduction among these groups. Neutrophil infiltration into pulmonary interstitial fluid could result in even further pulmonary edema and deteriorating inflammation[49,50]. Thus, we additionally evaluated pulmonary neutrophil infiltration. The neutrophils were labeled with PE-labeled anti-Ly-6G/Ly-6C and FITC-labeled anti-CD11b antibodies. The percentages of neutrophils in total cells were determined by flow cytometry. It was observed that neutrophils accounted for ~85% and ~75% of the total cells in KPN-infected and TRKP-infected mice, respectively (Fig. 7F). The percentages of neutrophils were apparently decreased after administration of TIG, TCT, TTCT, and Ts-TPGS/Cap nanorods, clearly indicating that pulmonary neutrophil infiltration was ameliorated. It is worth mentioning that TTCT treatment still exhibited the best effects among the treatments. Moreover, Ts-TPGS/Cap nanorods displayed an inhibitory effect on pulmonary neutrophil infiltration similar to that of TIG (15 mg/kg) in TRKP-infected mice. Representative images of flow cytometry were shown in Fig. 7G.

CRP is a well-documented indicator of bacterial infection[51]. The levels of CRP in BALF were determined using a commercial ELISA kit. As shown in Fig. 7H, the levels of CRP were markedly elevated in KPN- and TRKP-infected mice. However, its level was apparently decreased after treatment with TIG, TCT, and Ts-TPGS/Cap nanorods, and the effect was further enhanced by TTCT. In addition, BALF was utilized for bacterial colony analysis, and TTCT showed the lowest bacterial colonies in pneumonia mice infected by KPN and TRKP among the mice receiving various therapy (Fig. 7I and quantitative analysis results shown in supplementary Fig. 3). Overall, TTCT therapy consistently displayed the best therapeutic effect, with lower counts of WBCs and neutrophils in blood, lower pulmonary neutrophil infiltration, and higher reductions in CRP levels and bacterial colonies. These results could possibly be explained by antibacterial activity of Ts peptides and inhibitory effect of efflux pumps, thereby increasing the TIG concentration in the lungs and bacteria.

**Positron emission tomography-computed tomography (PET-CT) imaging**. PET/CT imaging systems have been widely used in clinical diagnosis and treatment[52]. Increased capillary permeability at the initial stage of infection could lead to the accumulation and activation of inflammatory cells, and activated inflammatory cells mainly metabolize glucose as an energy source[53]. Therefore, to provide more evidence of the antibacterial effect, PET imaging with [18]F-labeled fluorodeoxyglucose ([18]F-FDG) was employed to evaluate inflammation and infection in different groups. As shown in Fig. 8A, KPN and TRKP infection caused a significant increase in [18]FDG uptake in the lung. The levels of [18]FDG uptake were reduced after the infected mice were treated with TIG, TTCT, and Ts-TPGS/Cap, as reflected by decreased standardized uptake values (SUVs). This phenomenon

was further ameliorated by TTCT intervention, demonstrating that TTCT exhibited a superior effect compared with the other treatment groups, which was consistent with the results displayed in Fig. 7.

**Histopathology**. To evaluate the effects of different treatments on pathological changes, H&E staining was performed at 5 days post-administration. As expected, the lungs of the KPN- and TRKP-infected mice displayed typical pathological characteristics, such as alveolar wall thickening, edema, and leukocyte infiltration (Fig. 8B). Histological improvements were apparently observed after receiving TIG, TCT, and Ts-TPGS/Cap nanorods. TTCT displayed more pronounced amelioration of pulmonary inflammatory infiltration, edema, and alveolar wall thickening than the other groups. Furthermore, lungs were stained with myeloperoxidase (MPO) antibody to investigate neutrophil infiltration. Confocal images showed that bacterial infection resulted in significant neutrophil infiltration in the lungs, and this phenomenon was obviously ameliorated after the different treatments (Fig. 8C). Consistent with the results shown in Fig. 8A, TTCT still demonstrated better therapeutic effects than the other groups.

**Biosafety of TTCT**. The cytotoxicity of TPGS/Cap and Ts-TPGS/Cap nanorods, a critical factor for their further application in vivo, was evaluated by the MTT method in HUVECs and EA.hy926 cells. The cell viability was over 90% when the concentration of nanorods reached 0.4 mg/mL, implying that both TPGS/Cap and Ts-TPGS/Cap had excellent biocompatibility (Fig. 9A, B). The hemolysis percentage was ~ 3% for Ts, TPGS/Cap, and Ts-TPGS/Cap nanorods. For TIG, TCT, and TTCT, the hemolysis percentages were $6.80 \pm 0.11\%$, $3.92 \pm 0.52\%$, and $4.13 \pm 0.96\%$, respectively, implying that TIG encapsulation increased its biosafety (Fig. 9C). When the pneumonia mice caused by KPN and TRKP received TIG therapy, apparent fatty and hydropic degeneration in the liver was observed (Fig. 9D). However, no obvious pathologic changes in the collected organs were observed in the TCT, TTCT, and Ts-TPGS/Cap nanorod groups, indicating that the adverse effect of TIG was decreased after encapsulation into nanodrug delivery systems. These results were consistent with those shown in Fig. 9C.

**Discussion**

In summary, a TPGS-based and Ts-modified nanodrug delivery system with LPS targeting and synergistic antibacterial activity was designed for TIG delivery and therapy of acute pneumonia caused by MDR bacteria. The prepared Ts-TPGS/Cap nanorods could effectively encapsulate TIG and achieve sustained drug release. Through the binding between Ts and LPS, Ts-TPGS/Cap exhibited targeting and enhanced accumulation in both KPN and TRKP. TPGS could exert its inhibitory capacity on the activity of efflux pumps and the expression of *acrA*, *acrB* and *ramA* in

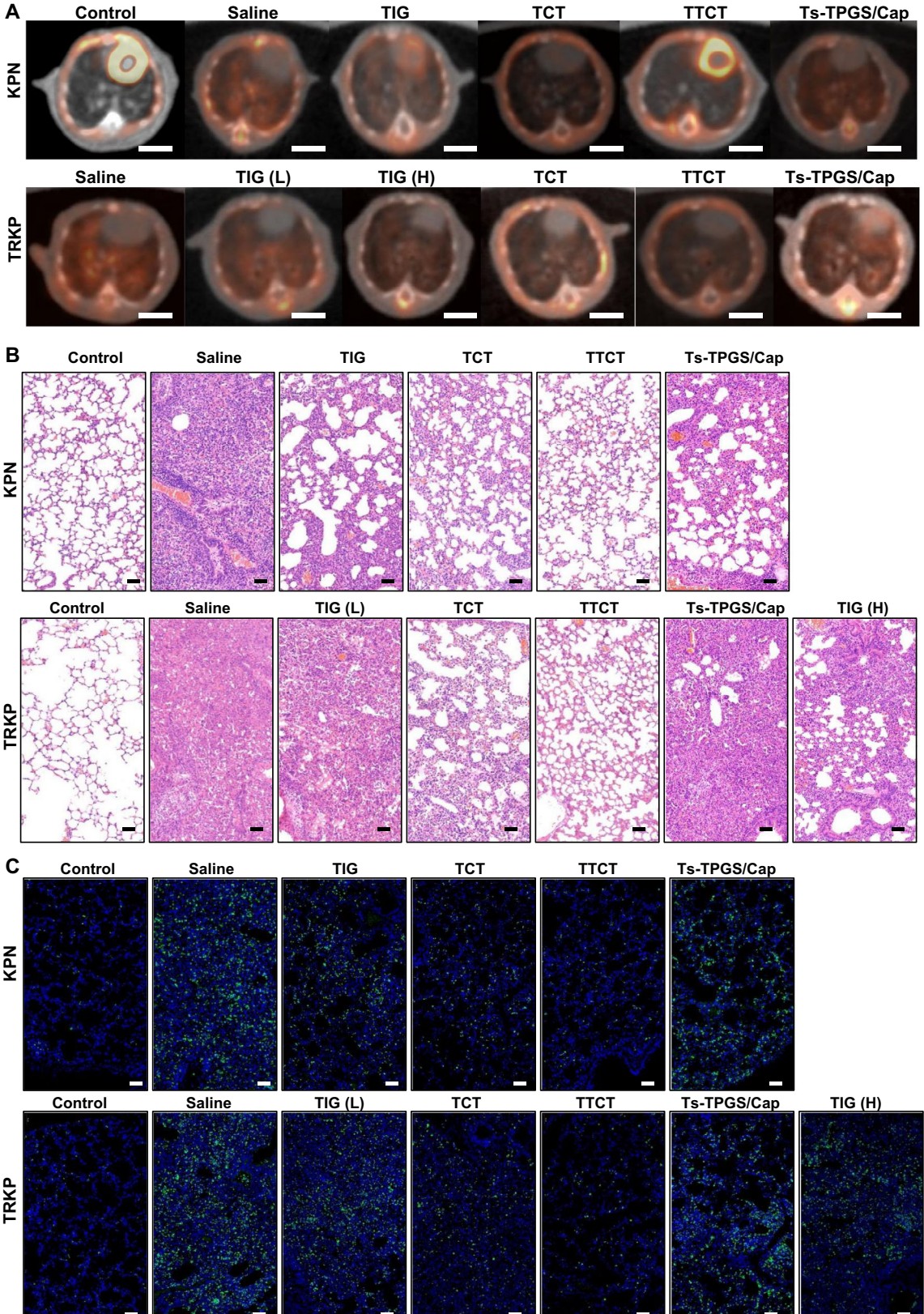

**Fig. 8 Histopathology observation of lung tissue. A** Representative images of the lungs observed by PET-CT after administration to various groups (Scale bar = 5 mm). **B** Representative images of lung tissue from different groups after H&E staining (scale bar = 50 μm). **C** Fluorescent images of the lung tissue collected from mice with pneumonia and stained with anti-MPO antibody, in which the green channel indicates neutrophil infiltration and the blue channel represents the nucleus. Scale bar = 50 μm. TIG free tigecycline, TIG(L) low dose of tigecycline (15 mg/kg), TIG (H) high dose of tigecycline (45 mg/kg), TCT tigecycline loaded TPGS/Cap nanorods, TTCT tigecycline-loaded Ts-TPGS/Cap nanorods, KPN *Klebsiella pneumonia*, TRKP tigecycline-resistant *Klebsiella pneumonia*. The experiments in **B**, **C** were repeated independently for three times with similar results.

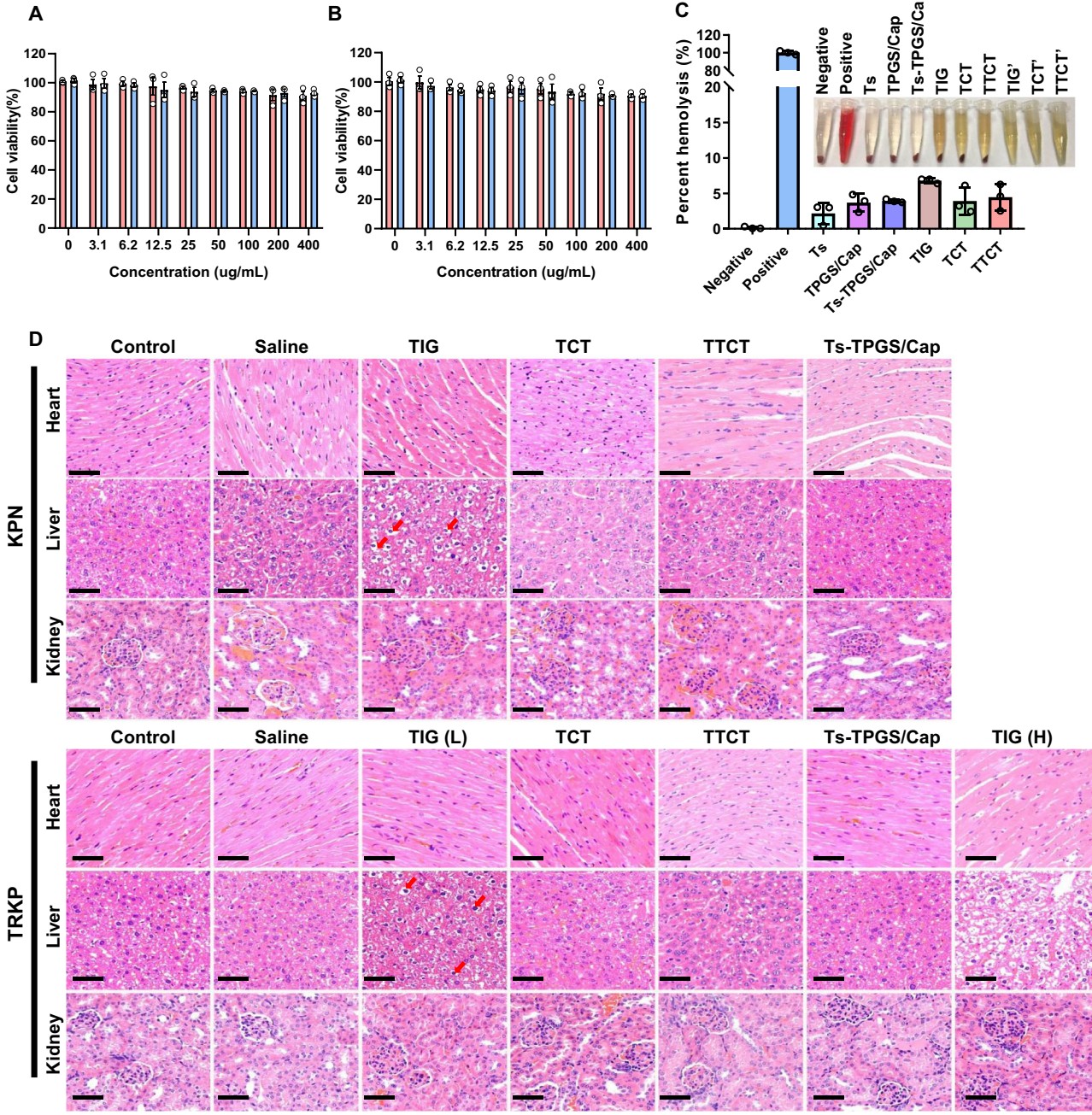

**Fig. 9 Evaluation of the biosafety of different treatments.** Cell viabilities of (**A**) HUVECs and **B** EA.hy926 cells after incubation with TPGS/Cap and Ts-TPGS/Cap at various concentrations for 48 h. Data are presented as the mean ± SD ($n = 3$ independent experiments). **C** In vitro hemolysis study after incubation with different preparations. Representative images of hemolysis of different preparations are shown, in which TIG', TCT' and TTCT' represent preparations without incubation with red blood cells. Data are presented as the mean ± SD ($n = 3$ independent experiments). **D** Histopathology images of the heart, liver, and kidney obtained from each group at 5 days post-injection, and the red arrow indicates physalides in liver (Scale bar = 50 μm). TIG free tigecycline, TIG(L) low dose of tigecycline (15 mg/kg), TIG (H) high dose of tigecycline (45 mg/kg), TCT tigecycline loaded TPGS/Cap nanorods, TTCT tigecycline-loaded Ts-TPGS/Cap nanorods, KPN *Klebsiella pneumonia*, TRKP tigecycline-resistant *Klebsiella pneumonia*. The experiments in D were repeated independently for three times with similar results. Source data are provided as a Source Data file.

TRKP. In this way, the TIG concentration inside bacteria was significantly higher in the TTCT group than in the TIG and TTCT groups. The synergistic antibacterial capacity between Ts and TIG further enhanced the antibacterial activity TTCT, thus overcoming the drug resistance of TRKP.

In mice with pneumonia, Ts-TPGS/Cap specifically accumulated in the lungs. TTCT administration could significantly reduce the WBC and neutrophil counts in blood samples and decrease the total cell and CRP levels in BALF. Moreover, TTCT

was capable of ameliorating neutrophil infiltration in the lungs and reducing bacterial colonies from BALF, thus apparently increasing the survival rates of mice with pneumonia caused by both KPN and TRKP. In addition, TTCT led to little toxicity to the liver. These results indicated that TTCT could overcome the drug resistance of TRKP.

Except for the classical *K. pneumoniae* (cKp) we focused on, hypervirulent *K. pneumoniae* (hvKp), which is more virulent than cKp, has emerged as a concerning global pathogen, and is capable

of causing community-acquired infections, often in healthy individuals[9]. It was reported that similar to cKp, hvKp strains are becoming increasingly resistant to antimicrobials via acquisition of mobile elements carrying resistance determinants. When extensively drug-resistant cKp strains acquire hvKp-specific virulence determinants, new hvKp strains emerge and result in nosocomial infection[10]. Considering the potential severity of hvKp infection and its propensity for metastatic spread, tigecycline plays important roles in the empirical treatment of carbapenem-resistant hvKp strains. However, tigecycline resistance has been reported for an cKp strain that evolved into an hvKP strain via acquisition of a portion of an hvKp virulence plasmid. Overexpression of the efflux pump gene *acrR* and its regulatory gene *ramA* has been demonstrated to be associated with tigecycline resistance[54]. Based on our data, we speculate that TTCT system still has advantages against carbapenem-resistant hvKp strains. For tigecycline-sensitive hvKp, TTCT exerts more than one distinct mechanism and the synergistic/addition activity between Ts and TIG might reduce the likelihood of bacteria developing resistance.

As a time-dependent antimicrobial agent with delayed post-antibiotic effect (PAE), the pharmacokinetics/pharmacodynamics (PK/PD) parameter associated with the clinical efficacy of tigecycline is AUC/MIC. For tigecycline-resistant hvKp, the increased MIC and the limited tigecycline dosage were the main reasons for the poor clinical efficacy[55]. The MIC reduction endow by TTCT would be beneficial for maximizing the efficacy and minimizing the drug-related adverse events of tigecycline.

In conclusion, TTCT might be a promising therapeutic candidate for infectious diseases caused by MDR gram-negative bacteria. However, the detailed mechanism underlying the binding between LPS and the Ts peptide requires further investigation.

## Methods

The research complied with all relevant ethical regulations and got approval from Zhejiang University. The in vivo study was approved by the Animal Care and Use Committee of the First Affiliated Hospital, College of Medicine, Zhejiang University.

**Materials**. The D-α-tocopherol polyethylene glycol 1000 succinate (TPGS1000) was provided by Sigma Aldrich Co., Ltd. (USA). The di-tert-butyl dicarbonate ((Boc)₂O) was purchased from Shanghai Medpep (Shanghai, China). The S-thanatin (Ts) peptide (GSKKPVPIIYCNRRSGKCQRM) and non-targeting peptide (GSKKPVPIIYANRRSGKAQRM) were synthesized by Guangzhou Sinoasis Pharmaceuticals Inc. (Guangzhou, China). Anhydrous calcium chloride, disodium hydrogen phosphate dodecahydrate, sodium citrate, sodium hydroxide, 2.5% glutaraldehyde solution and hydrochloric acid were obtained from Sinopharm Chemical Reagent Co., Ltd. (China). 6-diamidino-2-phenylindole (DAPI) and 3-(4,5-Dimethyl-thiazol-2-yl)−2,5-diphenyltetrazoli-nbromide (MTT) was supplied by Sigma Chem. Co., Ltd (USA). The KPN and TRKP strains we used were clinical isolates and kindly provided by State Key Laboratory for Diagnosis and Treatment of Infectious Diseases, The First Affiliated Hospital, College of Medicine, Zhejiang University (Hangzhou, China). Tigecycline (TIG) was supplied by Aladdin (Shanghai, China). Internal standard of tigecycline (D9-TIG) was supplied by J&K Chemical Co., Ltd. (Beijing, China). Nutrient agar and LB broth were provided by Hangzhou Microbial Reagent Co., Ltd. (Hangzhou, China). Fluorescein iso-thiocyanate (FITC), indocyanine green (ICG) and 4% paraformaldehyde (PFA) were brought from Meilun Biotechnology Co., Ltd. (Dalian, China). Fetal bovine serum (FBS) was obtained from Sijiqing Biological Engineering Materials Co. Ltd. (Hangzhou, China). Ethidium bromide (EB) and carbonyl cyanide m-chlorophenylhydrazone (CCCP) was obtained from Sigma Aldrich Co., Ltd. (USA). Dulbecco's Modified Eagle's Medium (DMEM) was supplied by Corning (USA). The oligonucleotide primers of *acrA*, *acrB*, *ramA* and 16S rRNA employed for real-time RT-PCR were synthesized by Sangon Biotech (Shanghai) Co., Ltd. PrimeScript™ RT reagent Kit and SYBR Premix Ex Taq Kit were purchased from Takara Biomedical Technology Co., Ltd. (Beijing, China). Anti-myeloperoxidase (MPO) antibody was purchased from Abcam (UK). CD11b Monoclonal Antibody (M1/70) and Ly-6G/Ly-6C Monoclonal Antibody (RB6-8C5) were purchased from eBioscience. The ELISA kits of CRP were obtained from Boster Biological Technology Co., Ltd (Wuhan, China). All other chemical reagents were of analytical or chromatographic grade.

**Cell lines and animals**. EA.hy926 (catalog number: GNHu39) was obtained from Chinese Academy of Sciences Cell Bank (Shanghai, China). Human umbilical vein endothelial cells (HUVECs) was provided by Cell Resource Center,IBMS, CAMS/PUMC (catalog number: 1101HUM-PUMC000437)

Male ICR mice (6 to 8 weeks old, 22–25 g) were provided by Zhejiang Medical Animal Center and had free access to food and water. Animals were housed at approximately 22 ± 2°C, humidity 50 ± 10% on a 12 h light/ 12 h dark cycle. The surgical procedures and in vivo experiments were performed according to the National Institutes of Health Guide for the Care and Use of Laboratory Animals and were approved by the Animal Care and Use Committee of the First Affiliated Hospital, College of Medicine, Zhejiang University (Reference number: 2019-227).

**Synthesis and characterization of Ts-TPGS**. Ts-TPGS (TT) was synthesized via an esterification reaction according to a previously reported method, as shown in Fig. 2A[56]. Briefly, Ts peptide (20 mg) and (Boc)₂O (11.2 μL) were added to 2 mL anhydrous DMSO in an ice bath, followed by stirring at room temperature for 12 h Then, 5 mL DMSO containing TPGS, DCC, and DMAP was added to the reaction system, and stirring was continued at 60 °C for 24 h. The Boc protecting group was removed with 2 M HCl. Subsequently, the reaction solution was placed into a 1.0 kDa MWCO dialysis bag and dialyzed by de-ionized water for 2 days. Then, the TT conjugate was obtained after lyophilization. ¹H-NMR was used to confirm the structures of Ts, TPGS, and TT.

**Preparation and characterization of Ts-TPGS/Cap nanorods**. Ts-TPGS/Cap were prepared by a hydrothermal method[57]. Briefly, 1 mL of Na₂HPO₄.12H₂O (30 mM) was added dropwise into a mixed solution consisting of 1 mL of CaCl₂ solution (50 mM) and 1 mL of TT solution (5 mg/mL). Then, 10 mg sodium citrate was added to the reaction system, and the pH was adjusted to 8 with NaOH. The resulting mixture was stirred at 100 °C for 4 h in a water bath, followed by dialysis against distilled water for 24 h. Then, Ts-TPGS/Cap nanorods were obtained after lyophilization. The TPGS/Cap nanorod was prepared by the same method. The particle size, size distribution, and zeta potential of the prepared nanorods were determined using a Zetasizer (3000HS, Malvern Instruments Ltd., UK). The morphology was observed by transmission electron microscopy (TEM, JEOL JEM-1230, Japan). The crystallinity of the prepared Ts-TPGS/Cap nanorods was determined by small-angle XRD.

**Preparation and characterization of the TTCT nanodrug delivery system**. TIG encapsulation was performed by adding a TIG/H₂O solution (2 mg/mL) dropwise to an aqueous Ts-TPGS/Cap solution (2 mg/mL) under constant stirring. After stirring for another 2 h, the mixed solution underwent dialysis (MWCO: 3.50 kDa) against DI water for 4 h. After centrifugation at 3500 g for 10 min, a TTCT nanodrug delivery system was obtained. The particle size, size distribution, and zeta potential were determined using a Zetasizer, and the morphology was examined by TEM. The stability of the nanorods was evaluated by the changes in particle sizes and PDI in both water and 10% FBS-containing PBS. The total drug content of TIG in the TTCT suspension and the unencapsulated TIG dissolved in water separated by an ultrafiltration tube (3.5 kDa, Millipore, Massachusetts, USA) were detected by a UV-vis spectrophotometer (TU-1080, Beijing, China) with a wavelength set at 245 nm. Then, the following equations were applied to determine the drug loading (DL) and encapsulation efficiency (EE).

$$DL\% = \frac{\text{mass of TIG encapsulated in TTCT}}{\text{mass of TTCT}} \times 100\% \qquad (1)$$

$$EE\% = \frac{\text{mass of TIG encapsulated in TTCT}}{\text{mass of TIG added}} \times 100\% \qquad (2)$$

The in vitro drug release behavior of TTCT nanodrug delivery system was evaluated by the dialysis method[58]. Briefly, 1 mL of TTCT solution was added to a dialysis bag (MW: 3.5 kDa) and dialyzed against PBS containing FBS (pH 7.4) under continuously shaking (60 rpm) at 37 °C. At predetermined time points, a fresh buffer solution was added after 0.1 mL collection of release medium. Then, HPLC-MS analysis was performed for determination of TIG concentration, and D9-TIG was employed as an internal standard (IS). Briefly, the collected samples were diluted 10-fold with blank matrix, followed by addition of 0.3 mL methanol containing IS (500 ng/mL). After vortex for 3 min, the mixture was centrifuged at 2500 g for 12 min at 4 °C, and the obtained supernatants were used for TIG concentration determination by HPLC-MS. Each measurement was performed in triplicate.

Chromatographic separation was performed on a Waters BEH-C18 column (50 mm × 2.1 mm, 1.7 μm). The temperature of the column was set at 35 °C. Gradient elution with a constant flow rate of 0.50 mL/min was performed. Mobile phase A consisted of 0.1% formic acid (v/v) and 10 mM ammonium formate in water, and mobile phase B was methanol. The gradient was set as follows: initiate at 5% B, equilibrate for 0.3 min, increase to 95% B over the next 1.7 min, maintain for 0.5 min, change back to 5% B, and finally re-equilibrate for 1.5 min prior to the next injection. The total run time was 4.0 min, and the injection volume was 2 μL. Mass spectrometry was performed in positive ESI mode, and the capillary voltage was 2500 kV. The collision energies were optimized at 64 V for TIG and 54 V for IS. The cracking voltage was 30 V for TIG and 32 V for IS. Quantification analysis

was conducted using multiple reaction monitoring (MRM). The protonated precursor → product ion combinations m/z were $586.31 \to 456.21$ for TIG and $595.30 \to 514.30$ for IS[59].

**Antibacterial activity of TTCT in vitro**. The in vitro antibiotic efficacy of the TTCT nanodrug delivery system against KPN and TRKP was evaluated by measuring the MICs and growth curves. MIC was the lowest concentration of the agents that inhibited visible bacterial growth. The determination of MIC values of various formulations (TIG, TCT, TTCT, Ts peptide, Ts-TPGS/Cap nanorods) was achieved via the microplate broth dilution method. The bacteria obtained in the mid-exponential growth phase was diluted to $5 \times 10^5$ CFU/mL ($5 \times 10^4$ CFU/well) and incubation with the formulations for 20 h for MIC tests. As for growth curve analyses, KPN and TRKP ($5 \times 10^5$ CFU/mL) were incubated with TIG, Ts peptide, TCT, Ts-TPGS/Cap, and TTCT. At predetermined time points, the absorbance of these cultures at a wavelength of 620 nm was detected, and the corresponding bacterial growth curves were plotted. The experiments were repeated three times.

SEM was used to observe the morphologies of bacteria receiving different treatments. Briefly, cultures of KPN and TRKP ($1 \times 10^8$ CFU/mL) were incubated with TIG, TCT, TTCT, Ts peptide, and Ts-TPGS/Cap nanorods for 4 h at 37 °C. The corresponding TIG and Ts concentrations were 2 µg/mL and 32 µg/mL, respectively. Then, the suspensions were centrifuged (2500 g, 10 min) for collection of the bacteria. The samples were fixed with 2.5% glutaraldehyde solution, washed with PBS, dehydrated with ethanol, dried under vacuum, coated with platinum, and then imaged by SEM (SEM; Hitachi SU-8010, Japan).

**Targeting efficacy of Ts-TPGS/Cap nanorod in vitro**. Fluorescein isothiocyanate (FITC) dissolved in ethanol (2 mg/mL) was added dropwise to Ts-TPGS/Cap solution (2 mg/mL). After stirring at room temperature for 2 h and dialyzing against deionized water for 6 h, FITC-labeled nanorods were obtained for further use. FITC-labeled NT-TPGS/Cap nanorods were prepared using the same method. Cultures of KPN and TRKP ($1 \times 10^8$ CFU/mL) were incubated with 0.1 mg/mL FITC-labeled Ts-TPGS/Cap or NT-TPGS/Cap nanorods for 2 h and 6 h, respectively. Then, the samples were harvested by centrifugation, stained with DAPI to label bacteria, washed with PBS, fixed with 4% paraformaldehyde (PFA), and imaged by confocal laser scanning microscopy (CLSM). Furthermore, the fluorescent signal inside bacteria was quantitatively detected by flow cytometry to investigate the targeting efficacy of Ts-TPGS/Cap nanorods.

**Activity of efflux pump evaluation**. To evaluate the activity of the efflux pump, EB and CCCP were employed as an efflux pump substrate and positive control, respectively[27]. Briefly, TRKP cultures ($1 \times 10^8$ CFU/mL) were incubated with different concentrations of TPGS and CCCP (100 µM) for 1 h, followed by incubation with EB for another 3 h for EB accumulation. Afterwards, the bacteria and supernatant were harvested by centrifugation (2500 g, 10 min). The fluorescent signal of EB accumulated inside bacteria was qualitatively observed by CLSM. The EB content in the supernatant was detected by a spectrofluorometer (λex = 530 nm; λem = 600 nm) for quantitative analysis of EB accumulation. Then, in comparison with the negative group, we calculated the increase of EB accumulation after bacteria pretreated by TPGS or CCCP.

**Real-time RT-PCR**. Real-time RT-PCR was used to investigate the levels of *acrA*, *acrB* and *ramA*, using 16 S rRNA gene as the reference. Briefly, KPN and TRKP cultures ($2 \times 10^8$ CFU/mL) were incubated with TPGS, Ts, Ts-TPGS and Ts-TPGS/Cap nanorods for 20 h at 37 °C (final TPGS and Ts concentrations at 0.1 mg/mL and 8 µg/mL, respectively). RNAiso Plus was applied to extract the DNase-treated RNA templates of the obtained bacterial cultures. After centrifugation, the concentrations of collected RNA were detected by Nanodrop spectrophotometer (Thermo Scientific, MA, USA). Subsequently, a PrimeScript RT-PCR kit (TaKaRa Bio) was utilized to reversely transcribe mRNA into cDNA. Afterwards, real-time PCR was conducted utilizing a SYBR Premix Ex Taq kit (TaKaRa Bio) on an Applied Biosystems StepOnePlus Real-Time PCR system. The $2-\Delta\Delta Cq$ method was used to estimate the relative expression of *acrA*, *acrB* and *ramA*. The details of all primer sequences are shown in Supplementary Table 2.

**TIG concentration inside bacteria**. We hypothesized that the targeting efficacy endowed by the Ts peptide and the inhibitory activity of the efflux pump endowed by TPGS would increase the TIG concentration inside bacteria and improve the antibacterial capacity of TIG. To confirm our hypothesis, the TIG concentration inside bacteria was detected by HPLC-MS, and D9-TIG was employed as an internal standard (IS). In brief, KPN and TRKP ($1 \times 10^8$ CFU/mL) were incubated with TIG, TCT, and TTCT (TIG concentration: 2 µg/mL) for 2 h and 6 h, respectively. Then, the bacteria were collected, washed with PBS, and lysed with 0.15 mL of methanol. After vortexing and centrifugation, a 0.1 mL aliquot of the samples was transferred to Eppendorf tubes, and 0.1 mL of IS methanol solution (50 ng/mL) was added and vortexed for 0.5 min. Then, the mixture was centrifuged at 2500 g for 12 min at 4 °C, and the obtained supernatants were used for TIG concentration determination by HPLC-MS.

**Fractional inhibitory concentration index testing**. The interaction between Ts and tigecycline was investigated by the micro-dilution checkerboard technique according to CLSI guidelines. Briefly, twofold dilutions of TIG and Ts were prepared. A 96-well microtiter plate was then filled with 50 µL of the appropriate concentrations of TIG solution, and this step repeated with Ts (total final volume of 0.1 mL). A bacterial suspension containing either KPN or TRKP was then dispensed into the wells with final bacterial concentration of ~$5 \times 10^4$ CFU/well. The MIC referred to the lowest concentration of an agent that could inhibit the bacterial growth determined by both visual reading and OD600 values. Following incubation, the synergistic/additive effect was determined by calculating the fractional inhibitory concentration index according to the following equation:

$$FICI = \frac{\text{MIC of A in combination}}{\text{MIC of A}} + \frac{\text{MIC of B in combination}}{\text{MIC of B}} \qquad (3)$$

The synergy or additive was defined according to standard criteria (FICI ≤ 0.5 was defined as synergistic; 0.5 < FICI ≤ 1 was defined as additive; 1 < FICI ≤ 4 was defined as indifference; FICI > 4 was defined as antagonism)[42].

**Cellular uptake**. EA.hy926 cells seeded in 24 well plates were allowed to adhere overnight. After incubation with 50 µg of FITC-labelled NT-TPGS/Cap or Ts-TPGS/Cap nanorods for different times, the cell nuclei were stained with DAPI. Then, cells were washed thrice with PBS, fixed with 4% paraformaldehyde, and observed by confocal laser scanning microscopy.

**Biodistribution in vivo**. Acute pneumonia mice model caused by KPN and TRKP was conducted by tracheal injection of 30 µL KPN or TRKP bacteria into the lungs of healthy mice ($1 \times 10^9$ CFU/mL) into the lungs of mice (25 ± 3 g). To observe the biodistribution, ICG-labeled Ts-TPGS/Cap nanorods were first prepared using a process similar to that of drug loading. At 5 h postoperaion, the mice were intravenously injected with ICG-labeled nanorods. Then, the mice were sacrificed, and the organs (heart, liver, spleen, lung, and kidney) were collected 5 h and 24 h after administration. An IVIS® spectrum system (Caliper, Hopkinton, MA, USA) was applied for qualitative observation and semi-quantitative analysis of the the accumulation of the fluorescent signal inside the tissue.

**Experimental design**. The KPN-infected mice with pneumonia were randomly divided into five groups ($n = 6$). Saline, TIG, TCT, TTCT, and Ts-TPGS/Cap (TIG dose: 15 mg/kg, nanorod dose: 100 mg/kg) were intravenously injected 5 h after modeling. The TRKP-infected mice with pneumonia were randomly divided into six groups and intravenously administered saline, TIG (low dose: 15 mg/kg, high dose: 45 mg/kg), TCT (TIG: 15 mg/kg), TTCT (TIG: 15 mg/kg), and Ts-TPGS/Cap nanorods. Normal healthy mice injected with 30 µL of saline into the lungs via the trachea were used as controls.

**Anti-infective efficacy of TTCT in vivo**. Recording of a five-day survival rate of the mice receiving different therapy was conducted for preliminary evaluation the anti-infective efficacy of TTCT in vivo. Blood specimens were collected at 24 h after the various treatments, and hematology analysis was performed by an automated Beckman Analyzer (Beckman Instruments GmbH, Munich, Germany). For further evaluation of in vivo anti-infective efficacy, BALF was collected after 24 h according to a method reported previously. In brief, tracheal intubation was primarily established accompanied by a ligature on the trachea by surgical lines. Cold PBS was injected via the endotracheal tube and gently withdrawn five times. A total of 2 mL PBS was used, and the recovery of BALF reached 80%. Then, after centrifuging the BALF (100 g, 10 min), the supernatant and the cell pellet were harvested.

The CRP levels in the supernatant were determined by enzyme-linked immune sorbent assay (ELISA). Serial dilutions of the supernatant obtained from BALF were cultured in MH agar plates, and the number of colonies was counted. The cell pellet was treated with ACK lysis buffer to remove the red blood cells, and the total number of cells was determined by flow cytometry. Furthermore, after staining with PE-labeled anti-Ly-6G/Ly-6C (Cat#: 12-5931-82, 0.03 µg/test, eBioscience™) and FITC-labeled anti-CD11b (Cat#: 11-0112-41, 0.25 µg/test, eBioscience™) antibodies, the ratios of neutrophils in total cells were detected by flow cytometry[49]. The data were analyzed by FlowJo software.

For pathological observation, the lungs in each group were harvested, fixed with 4% PFA for 2 days, embedded in paraffin, sectioned at a thickness of 4 µm, and mounted onto glass slides. Then, the sections were stained with hematoxylin and eosin (H&E) and imaged using an optical microscope. In addition, the sections were stained with primary anti-MPO antibody (Cat#: ab208670, 1:500, Abcam, UK) and the appropriate secondary antibody. After staining with DAPI to label the nuclei, the sections were imaged by CLSM.

**PET-CT imaging**. ¹⁸F-FDG was synthesized and kindly provided by the PET Centre of our hospital. The animals were fasted overnight before the experiment. ¹⁸F-FDG (250 µCi in 0.2 mL) was intravenously injected 0.5 h prior to PET scanning. Then, the mice were anesthetized and placed prone in the center of a Siemens Inveon combined micro PET-CT scanner (Siemens Preclinical Solution USA, Inc., Knoxville, TN, USA) with limbs stretched. MicroCT scanning was

conducted with the following parameters: 80-kV X-ray tube voltage, 500-μA source current, 120-ms exposure time, and 120 rotation steps. A 10-min PET static acquisition was performed, and the corresponding images were reconstructed by the OSEM (ordered set expectation maximization) algorithm for 3D PET reconstruction. Inveon Research Workplace 4.1 (Siemens, Erlangen, Germany) was used to analyze the acquired images. The standardized uptake value (SUV, the unit of SUV is g/ml) was calculated by the formula:

$$SUV = \frac{RTA/cm^3}{RID} \times BW \qquad (4)$$

RTA represents the measured radiotracer tissue activity (mCi), RID refers to the radiotracer injected dose (mCi), and BW is the body weight (g) of the model mouse. The maximum SUV (SUV$_{max}$) in the lung was recorded.

**Biosafety of TTCT**. The cytotoxicity of Ts-TPGS/Cap and TPGS/Cap nanorods against HUVECs and EA.hy926 was evaluated by MTT method. Briefly, cells seeded into 96-well plates at a density of $1 \times 10^4$ cells/well were incubated with various concentrations of Ts-TPGS/Cap and TPGS/Cap nanorods. After a 48 h incubation period, 20 μL MTT aqueous solution (5 mg/mL) was added and then incubated for another 4 h. After that, the medium was replaced with 100 μL DMSO, and the absorbance of each well was recorded using a Bio-Rad 680 microplate reader at a wavelength of 570 nm. Cell viability was calculated in reference to negative cells without exposure to test agents.

An in vitro hemolysis assay was performed to investigate the biosafety of the nanodrug delivery system. First, fresh rat blood samples were collected from the orbit and stabilized with ethylene-diamine tetraacetic acid. Then, red blood cells (RBCs) were obtained after centrifugation (100 $g$, 10 min), washed with saline five times, and diluted to a 2% RBC suspension. Next, 0.5 mL of the RBC suspension was mixed with 0.5 mL of Ts peptide, TPGS/Cap, Ts-TPGS/Cap, TIG, TCT, and TTCT. Saline was used as a negative control, and pure water was used as a positive control. In addition, 0.5 mL of TIG, TCT, and TTCT mixed with saline served as the sample control. All the samples were kept at room temperature for 4 h. Next, the mixtures were centrifuged (100 $g$, 10 min), and the absorbance of the supernatant at a wavelength of 541 nm was detected by an ultraviolet spectrophotometer. Finally, the hemolysis percentages were calculated by the following formula:

$$Percent\ hemolysis\ (\%) = \frac{A_{sample} - A_{negative}}{A_{positive} - A_{negative}} \times 100\% \qquad (5)$$

To further investigate the biosafety of the various preparations in vivo, tissues, including heart, liver, lung and kidney, were harvested after the mice were sacrificed. The tissues were stained with H&E and imaged using an optical microscope.

**Statistical analysis**. All the values are expressed as the mean ± SD, as described in the figure legends. A comparative analysis of the difference between groups was conducted using one-way analysis of variance (ANOVA) with post hoc Tukey tests using SPSS 22.0 (95% confidence interval). Differences were considered statistically significant when the $p$ value was less than 0.05.

**Reporting summary**. Further information on research design is available in the Nature Research Reporting Summary linked to this article.

## Data availability

Data are available within the Article, Supplementary Information or from the corresponding author upon reasonable request. Source data are provided with this paper.

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

## Acknowledgements

X.Y.L. acknowledges the National Natural Science Foundation of China (81971981). X.J.W. acknowledges the National Natural Science Foundation of China (82003660) and the Nature Science Foundation of Zhejiang province (LQ20H300003 and LYY21H300004). S.P.J acknowledges the National Natural Science Foundation of China (82170725). N.C acknowledges the Nature Science Foundation of Zhejiang province (LYY18H310003).

## Author contributions

J.S.P., D.Y.Z., and L.X.Y. conceived the idea and provided guidance. W.X.J. and X.X.L. prepared the samples and conducted the experiment. W.X.J., Z.S.J., C.N, S.Y.F., M.K.F., H.D.S., and Li.L. discussed and analyzed the data. W.X.J. wrote the manuscript with assistance from J.S.P., D.Y.Z., and L.X.Y. D.Y.Z.provided all the experimental conditions. The authors were all involved in interpreting and constructing the figures.

## Competing interests

The authors declare no competing interests.
