## [Peer Review File · Nature Communications]

REVIEWER COMMENTS

Reviewer #1 (Remarks to the Author):

Tigecycline (TIG)-resistant *Klebsiella pneumoniae* is difficult to treat. The authors developed calcium phosphate (Cap) nanorods consisting of S-thanatin peptide (Ts)-conjugated tocopheryl polyethylene glycol succinate (TPGS) to encapsulate TIG, and applied the final product, Ts-TPGS/Cap/TIG (TTCT), to overcome therapy resistance. They described preparation and characterization of the drug delivery system, evaluated antibacterial activity in vitro and in vivo, explored mechanism of drug action, and assessed potential toxicity from TTCT. While the antibacterial activity seems convincing, the work lacks details and proper controls in key studies are missing.

1. There is no experiment to properly assign the roles from individual components in the nanorods on drug action. How are nanorods transported to the target organ? Are they internalized by cells in the lung? Which efflux pump is inhibited (AcrAB, OqxAB, or ramA)? By free TPGS, or free Ts-TPGS, or Ts-TPGS in nanorods?
2. Fig. 1. There are red circles and squares, but no text to explain them in the figure legend.
3. Fig 2. Over 50% drug molecules got released within the first hour in the in vitro drug release test (Fig. 2G). The result indicates that the majority of drug molecules will be released before the nanorods get accumulated in the lung. Do nanorods disassemble in circulation? This seems to be a design defect. In addition, there is no scale bar in Fig. 2D.
4. Fig. 4A-D. A non-targeting peptide-conjugated TPGS/Cap should serve as a proper control.
5. Fig. 4E. EB accumulation served as an indicator for drug accumulation. Is EB subjected to the same efflux pump(s) as TIG?
6. Fig. 7. Very fuzzy PET-CT images.

Reviewer #2 (Remarks to the Author):

The work titled "TPGS-based and S-thanatin functionalized nanorods for overcoming drug resistance in *Klebsiella pneumoniae*" showed as the use combined of Ts-TPGS/Cap nanocarrier produce targeted accumulation in bacteria of tigecycline and exhibited antibacterial capacity both in vitro and in vivo. The obtained TIG-loaded Ts-TPGS/Cap nanorods (TTCT) could enhance tigecycline accumulation in bacteria

via an inhibitory effect on efflux pumps and the targeting capacity to bacteria, thereby showing excellent antibacterial effects and overcoming the drug resistance of TRKP.

The work is very interesting but I have the following considerations to take in account the authors:

Major considerations

The study shows important results in relation a in vitro e in vivo experiments but must carry out more experiments about microbiology field to confirm the conclusion of the accumulation of tigecycline in the bacteria by inhibition of the efflux pumps through Ts-TPGS/Cap nanocarrier.

The authors must analyze the protein expression of the efflux pumps and the putative mechanisms of activity synergy by checkerboard assays (to calculate fractional inhibitory concentration index [FICI]) and Optical densities growth curves.

Moreover, the inhibitors of the efflux pumps could produce heteroresistance or resistance mechanisms associated to mutations in regulatory genes. It must be analyzed due to Ts-TPGS/Cap nanocarrier could not to success in the treatment of patients.

Minor considerations

-To improve the quality of the Abstract Figure

-To analyze the results in clinical strains with mechanisms of resistance involve the overexpression of the efflux pumps

-To study the results in clinical strains tigecycline-resistant by overexpression of the efflux pumps.

REVIEWER COMMENTS

Reviewer #1 (Remarks to the Author):

Tigecycline (TIG)-resistant *Klebsiella pneumoniae* is difficult to treat. The authors developed calcium phosphate (Cap) nanorods consisting of S-thanatin peptide (Ts)-conjugated tocopheryl polyethylene glycol succinate (TPGS) to encapsulate TIG, and applied the final product, Ts-TPGS/Cap/TIG (TTCT), to overcome therapy resistance. They described preparation and characterization of the drug delivery system, evaluated antibacterial activity in vitro and in vivo, explored mechanism of drug action, and assessed potential toxicity from TTCT. While the antibacterial activity seems convincing, the work lacks details and proper controls in key studies are missing.

1. There is no experiment to properly assign the roles from individual components in the nanorods on drug action. How are nanorods transported to the target organ? Are they internalized by cells in the lung? Which efflux pump is inhibited (AcrAB, OqxAB, or ramA)? By free TPGS, or free Ts-TPGS, or Ts-TPGS in nanorods?

Response: Thanks so much for your comment.

Firstly, the roles of the individual components are listed as following: (1) Calcium phosphate nanoparticles (Cap) were used as drug carriers for TIG encapsulation and nanodrug delivery system preparation. (2) The stability of Cap was quite poor¹, and TPGS was served as the stabilizer to improve the stability of Cap. As shown in Fig.2E-G, TPGS modified nanoparticles demonstrated excellent stability in both water and 10% fetal bovine serum (FBS)-containing PBS at pH 7.4. As previously reported, the small particle size of the nanodrug delivery system was beneficial for penetrating the cell walls². The particle size of the drug delivery system was ~25 nm (Fig. 2B), showing good bacterial penetrating activity (shown in Fig.4A). Additionally, TPGS could exert inhibitory effects on efflux pumps. Data in Fig.4E-F confirmed the inhibitory capacity of TPGS on activity of efflux pumps. Data in supplementary Fig.1 displayed the inhibitory effect of TPGS on expression levels of tigecycline-resistance related genes AcrA, AcrB and ramA. (3) S-thanatin (Ts) peptide is a short antimicrobial peptide that exhibits lipopolysaccharide (LPS) binding affinity. It acted

as a targeting peptide to increase the bacterial uptake and exerted synergy antibacterial activity with tigecycline. The results in Fig.4A-D confirmed its targeting capacity in vitro and the bio-distribution of Ts modified nanorods confirmed the targeting efficiency in vivo (Fig.5). Ts peptide demonstrated synergy antibacterial activity with tigecycline on TRKP (FICI = 0.325), and additive antibacterial activity on KPN (FICI = 0.75) (supplementary Table 1). The roles of the individual components in the nanorods have been clarified in the “Results” section of the revised manuscript.

Secondly, pneumonia may evoke disruption of pulmonary endothelial barrier integrity³, and the vascular permeability may be significantly increased at the sites of bacterial infection⁴. It has been reported that the enhanced permeability and retention (EPR) effect might operate to deliver nanoparticles to bacterial-infected tissues⁵. We speculated that the nanorods we prepared might be distributed into the infected lung via EPR effect after intravenous administration. Then the nanorods could be internalized by vascular endothelial cells (Data in supplementary Fig.2 demonstrated that the nanorods had excellent cellular internalization capacity) and be transported across the blood vessel. The binding affinity between Ts and LPS promoted the accumulation of Ts-functionalized nanorods in the infected lung (show in Fig.5).

Supplementary Fig.2 Fluorescence images of EA.hy926 cells after incubation with NT-TPGS/Cap and Ts-TPGS/Cap nanorods. The cells were labeled with DAPI (blue),

and the green fluorescence signal indicated the nanorods. Scale bar = 50 μm .

Thirdly, the mechanism of tigecycline resistance in TRKP is related to the high expression of RND-type efflux pump genes (AcrAB and OqxAB) and upstream binding site (ramA) ⁶. The expression levels of AcrA, AcrB and ramA genes were investigated by real-time reverse transcription-PCR (real-time RT-PCR) as you suggested. KPN and TRKP harvested in the mid-exponential growth phase were incubated with TPGS, Ts, Ts-TPGS and Ts-TPGS/Cap nanorods, respectively. As shown in supplementary Fig.1, expression levels of AcrA, AcrB, ramA were up-regulated significantly compared with those in KPN. Intervention of TPGS, Ts, Ts-TPGS and Ts-TPGS/Cap obviously down-regulated the expression levels. It is worth mentioning that Ts treatment also down-regulated the expression levels of efflux pumps, and we speculated that the phenomenon was associated with the Ts activity on the outer membrane of the bacteria ^{7, 8}. Ts-TPGS and Ts-TPGS/Cap nanorods exhibited the strongest efficiency in decreasing the expression levels of efflux pump genes, which could be explained by combinational suppression capacity of TPGS and Ts. Based on these results, we confirmed that TPGS had an inhibitory effect on the activity and expression levels of efflux pumps of TRKP. The results were consistent with those reported previously ⁹.

Supplementary Fig.1 The inhibitory effect of TPGS on efflux pumps. The expression levels of (A) AcrA, (B) AcrB and (C) ramA of TPGS, Ts, Ts-TPGS and Ts-TPGS/Cap nanorods treated TRKP by RT-PCR method. KPN and TRKP without any treatments were used as control. The error presents the standard deviation from the mean.

In addition, the expression levels of OqxA and OqxB were evaluated by real-time RT-PCR as well. No significant difference was observed between KPN and TRKP (data not shown). Thus, we speculated that OqxAB was possibly not related to the mechanisms of tigecycline resistance on the TRKP strain we used.

2. Fig. 1. There are red circles and squares, but no text to explain them in the figure legend.

Response: Thanks so much for your comment, and we were so sorry about the missing information. The red circles represented the characteristic peaks of the structure displayed in red squares. The figure legend of Fig.1 has been revised and the corresponding content has been added in the manuscript.

3. Fig 2. Over 50% drug molecules got released within the first hour in the in vitro drug release test (Fig. 2G). The result indicates that the majority of drug molecules will be released before the nanorods get accumulated in the lung. Do nanorods disassemble in circulation? This seems to be a design defect. In addition, there is no scale bar in Fig. 2D.

Response: Thanks so much for your comment.

Firstly, the drug release curves of TCT and TTCT were performed in PBS (pH = 7.4) and the amount of the release medium satisfied the sink condition. Secondly, the release media were totally replaced with fresh buffer solution at predetermined time points. As we know, sink condition was difficult to meet in vivo. Drug diffusion from nanoparticles would be slowed as concentration difference gets smaller. Additionally, there were many interfering factors of drug release in vivo, such as the protein levels. So, the release behavior confirmed the fact that the prepared nanodrug delivery systems had sustained drug release behavior compared with that of free drug, and Ts modification did not affect the drug release behavior.

In order to better simulate the drug release condition in vivo, the in vitro drug release behavior of the TTCT nanorods was evaluated by the dialysis method in 10% fetal bovine serum (FBS)-containing PBS at pH 7.4. The drug concentration and the

volume of the nanorods were consistent with those used in pharmacodynamic evaluation. At predetermined time points, 0.1 mL medium was collected and replaced with fresh buffer solution. Then, the drug concentration was detected by HPLC-MS. The results in revised Fig.2H displayed that ~30% drug molecules got released within the first hour. The accumulative release reached to ~50% at 8 h.

Fig.2H In vitro drug release profiles of TCT and TTCT in 10% fetal bovine serum (FBS)-containing PBS at pH 7.4 and 37 °C. Data are expressed as the mean \pm SD with n = 3 for each group.

Then, the stability of the nanorods in circulation were evaluated by the changes in particle sizes and PDI after incubation with PBS + 10% FBS. The data in Fig.2G demonstrated that the particle size and PDI remained almost constant with small changes within 24 h, indicating that TCT and TTCT would not disassemble in circulation.

Fig.2G Changes in particle size and PDI of TCT and TTCT in 10% fetal bovine serum (FBS)-containing PBS. Data are expressed as the mean \pm SD with $n = 3$ for each group.

Scale bar in Fig. 2D has been added in the revised manuscript.

4. Fig. 4A-D. A non-targeting peptide-conjugated TPGS/Cap should serve as a proper control.

Response: Thank you so much for your constructive suggestion. Thanatin (GSKKPVPIIYCNRRRTGKCQRM) is an inducible 21-residue insect peptide with a disulfide bond between Cys11 and Cys18^{10, 11}. Thanatin is characterized by a β -hairpin structure in its C-terminal region, which is considered integral for its activity¹². It was reported that the β -hairpin structure is stabilized by the disulfide bond¹³, and an analog of thanatin with the two Cys residues replaced by Ala was found to be largely inactive¹⁴, indicating that the disulfide bond is important for antibacterial activity. Considering this, an analog of Ts peptide with two Cys residues replaced by Ala (GSKKPVPIIYANRRSGKAQRM) was synthesized and served as the non-targeting peptide. The non-targeting peptide-conjugated TPGS/Cap (NT-TPGS/Cap) was prepared according to the preparation process of Ts-TPGS/Cap. The in vitro targeting efficiency of Ts-TPGS/Cap nanorods was reevaluated. The results were displayed in Fig.4A-D in the revised manuscript.

As shown in Fig. 4A, both NT-TPGS/Cap and Ts-TPGS/Cap nanorods displayed a gradual increase in the intro-bacteria fluorescent signal with a prolonged incubation time. Ts-TPGS/Cap nanorods displayed stronger bacterial internalization capacity than NT-TPGS/Cap nanorods in both KPN and TRKP at 2 and 6 h. The results were consistent with those obtained by flow cytometry (Fig. 4B and 4C). The fluorescent signals in KPN after incubation with Ts-TPGS/Cap nanorods were ~2-fold greater than those of NT-TPGS/Cap nanorods at both 2 h and 6 h (Fig. 4D). For TRKP, similar results were observed, suggesting that Ts-TPGS/Cap nanorods exhibited better bacterial internalization activity, which was possibly attributed to the targeting moiety Ts peptide.

Fig. 4 Investigation of the mechanisms by which TTCT overcomes drug resistance.

(A) Fluorescence images of KPN and TRKP after 2 h and 6 h of incubation with NT-TPGS/Cap and Ts-TPGS/Cap nanorods. The bacteria were labeled with DAPI (blue), and the green fluorescence signal indicated the nanorods. Scale bar = 20 μ m. Fluorescence within KPN (B) and TRKP (C) detected by flow cytometry after incubation with NT-TPGS/Cap and Ts-TPGS/Cap. (D) Quantitative analysis of fluorescence inside KPN and TRKP after coculture with nanorods. *, $p < 0.05$ indicates a significant difference existed between groups. Data are expressed as the mean \pm SD ($n = 3$). (E) Confocal fluorescent images of TRKP after exposure to EB

for 4 h, following pretreatment with TPGS and CCCP. Scale bar = 10 μm . (F) Quantitative analysis of the increase in EB accumulation compared with the negative control groups (n = 3). (G) TIG concentration detected by HPLC-MS inside KPN and TRKP after incubation with TIG-containing preparations (n = 5). *, p < 0.05 indicates a significant difference between groups. Data are expressed as the mean \pm SD.

5. Fig. 4E. EB accumulation served as an indicator for drug accumulation. Is EB subjected to the same efflux pump(s) as TIG?

Response: Thanks so much for your comment. The most common mechanism of tigeicycline resistance is the overproduction of **non-specific active** resistance-nodulation-cell division (RND) efflux pumps such as AcrAB-TolC and OqxAB¹⁵. RND family is a powerful efflux system with a broad spectrum of drug specificity¹⁶. Ethidium bromide (EB) is an efficient and common substrate for RND multidrug transporters as well.¹⁷ As a fluorescent substrate of RND pumps, EB diffuses passively through the bacterial membrane and becomes strongly fluorescent after its irreversible binding to nucleic acids. In the absence of efflux transporters, EB enters cells and accumulates within them via its intercalation into DNA. When active efflux transporters are expressed, fluorescence is suppressed¹⁸. Therefore, EB has been extensively used as a probe to evaluate the efflux activity of bacterial cells^{2,19}. It was reported that after *acrAB*-deficient *E. coli* incubation with EB, the fluorescence of EB inside-bacteria increased, indicating that EB is subjected to AcrB-AcrA efflux pump²⁰. Therefore, EB accumulation was served as an indicator for drug accumulation in our study.

6. Fig. 7. Very fuzzy PET-CT images.

Response: Thanks so much for your comment. We have tried our best to improve the quality of PET-CT images in the revised manuscript.

Reviewer #2 (Remarks to the Author):

The work titled “TPGS-based and S-thanatin functionalized nanorods for overcoming drug resistance in *Klebsiella pneumonia*” showed as the use combined of

Ts-TPGS/Cap nanocarrier produce targeted accumulation in bacteria of tigecycline and exhibited antibacterial capacity both in vitro and in vivo. The obtained TIG-loaded Ts-TPGS/Cap nanorods (TTCT) could enhance tigecycline accumulation in bacteria via an inhibitory effect on efflux pumps and the targeting capacity to bacteria, thereby showing excellent antibacterial effects and overcoming the drug resistance of TRKP.

The work is very interesting but I have the following considerations to take in account the authors:

Major considerations

The study shows important results in relation a in vitro e in vivo experiments but must carry out more experiments about microbiology field to confirm the conclusion of the accumulation of tigecycline in the bacteria by inhibition of the efflux pumps through Ts-TPGS/Cap nanocarrier.

1. The authors must analyze the protein expression of the efflux pumps and the putative mechanisms of activity synergy by checkerboard assays (to calculate fractional inhibitory concentration index [FICI]) and Optical densities growth curves.

Response: Thanks so much for your comment.

Firstly, the mechanism of tigecycline resistance in TRKP is related to the high expression of RND-type efflux pump genes (AcrAB and OqxAB) and upstream binding site (ramA) ⁶. The expression levels of AcrA, AcrB and ramA genes were investigated by real-time reverse transcription-PCR (real-time RT-PCR) as you suggested. KPN and TRKP harvested in the mid-exponential growth phase were incubated with TPGS, Ts, Ts-TPGS and Ts-TPGS/Cap nanorods, respectively. As shown in supplementary Fig.1 , expression levels of AcrA, AcrB, ramA were up-regulated significantly compared with those in KPN. Intervention of TPGS, Ts, Ts-TPGS and Ts-TPGS/Cap obviously down-regulated the expression levels. It is worth mentioning that Ts treatment also down-regulated the expression levels of efflux pumps, and we speculated that the phenomenon was associated with the Ts activity on the outer membrane of the bacteria ^{7, 8}. Ts-TPGS and Ts-TPGS/Cap

nanorods exhibited the strongest efficiency in decreasing the expression levels of efflux pump genes, which could be explained by combinational suppression capacity of TPGS and Ts. Based on these results, we confirmed that TPGS had an inhibitory effect on the activity and expression levels of efflux pumps of TRKP. The results were consistent with those reported previously⁹.

Secondly, the interaction between Ts and tigecycline was investigated by the micro-dilution checkerboard technique according to CLSI guidelines. Briefly, twofold dilutions of TIG and Ts were prepared. A 96-well microtiter plate was then filled with 50 μ L of the appropriate concentrations of TIG solution, and this step repeated with Ts (total final volume of 0.1 mL). A bacterial suspension containing either KPN or TRKP was then dispensed into the wells with final bacterial concentration of $\sim 5 \times 10^4$ CFU/mL. The MIC was determined as the lowest concentration of a drug that could inhibit the growth of microorganism by both visual reading and OD600 determined by a microtiter plate reader. Following incubation, the synergistic/additive effect was determined by calculating the fractional inhibitory concentration index according to the formula: $FICI = (\text{MIC of A in combination} / \text{MIC of A}) + (\text{MIC of B in combination} / \text{MIC of B})$. The synergy or additive was defined according to standard criteria ($FICI \leq 0.5$ was defined as synergistic; $0.5 < FICI \leq 1$ was defined as additive; $1 < FICI \leq 4$ was defined as indifference; $FICI > 4$ was defined as antagonism)⁸. The results were shown in supplementary Table 1. The data exhibited that the fractional inhibitory concentration indices between Ts and tigecycline against KPN and TRKP were 0.75 and 0.375, respectively, indicating that Ts exerted synergistic effect with tigecycline against TRKP and additive effect with tigecycline against KPN.

Supplementary Table 1. FIC indices against KPN and TRKP

Microorganisms	Combination	MIC (ug/mL)		FICI	Interpretation
		Individual	Combined		
TRKP	TIG	4	1	0.375	synergistic
	Ts	16	2		
KPN	TIG	1	0.5	0.75	additive

2. Moreover, the inhibitors of the efflux pumps could produce heteroresistance or resistance mechanisms associated to mutations in regulatory genes. It must be analyzed due to Ts-TPGS/Cap nanocarrier could not to success in the treatment of patients.

Response: Thank you so much for your constructive suggestion.

To address the concern, population analysis profiling (PAP) was used to detect the heteroresistance of TTCT referring to the method reported previously²¹. Briefly, 50 ul of TRKP was suspended in 5 ml Luria-Bertani broth medium and shaken at 185 rpm at 37°C overnight. Ten-fold serially diluted bacterial suspensions were prepared, and plated onto MH agar plates with TTCT concentrations in a 2-fold-change gradient (tigecycline concentration ranging from 1 to 8 ug/mL). After 24 h of incubation at 37°C, the number of colonies formed are counted to determine the frequency of resistant subpopulation. Tigecycline heteroresistance was defined as a population containing resistant subpopulations with MICs at least 8-fold higher than the original strain at frequencies of 10^{-6} to 10^{-7} ^{22, 23}. In our study, resistant subpopulations were observed, and the frequency at tigecycline concentration of 8 ug/mL was 2.78×10^{-7} , indicating the existence of heteroresistance.

To measure the stability of the resistant subpopulation, a single colony from the plate with tigecycline concentration of 8 ug/mL was selected and serially passaged without antibiotics for 7 days, then reanalyzed to determine the TTCT MIC²¹. We found that the resistance of the subpopulation reverts to susceptibility, indicating that the heteroresistance was unstable. Similar results also reported that heteroresistance was often lost after multiple generations in the absence of antibiotic selective pressure²⁴. We speculated that the unstable heteroresistance was possibly related to the fact that TTCT exerted more than one distinct mechanism and the synergistic antibacterial activity between Ts and TIG.

Despite of the existence of heteroresistance, our reasearch focused on enhancing

the antibacterial activity of tigecycline, overcoming the tigecycline resistance of TRKP and achieving effective therapy of pneumonia caused by TRKP. Our results demonstrated that the TTCT nanorods we constructed could realize significant MIC reduction from 4 ug/mL to 1 ug/mL on TRKP. Tigecycline is a time-dependent antimicrobial agent with long post-antibiotic effect (PAE), and its pharmacokinetics/pharmacodynamics (PK/PD) is assessed by AUC/MIC. The increase of MIC and the limiting tigecycline dosage were the main reasons for the poor clinical efficacy²⁵. The MIC reduction endowed by TTCT would be beneficial for maximizing the efficacy and minimizing the drug-related adverse events of tigecycline. The heteroresistance of TTCT would need further investigation in our future work. Thanks for your valuable and constructive comments.

Minor considerations

-To improve the quality of the Abstract Figure

Response: Thanks so much for your comment. Higher quality of the Abstract Figure was provided in the revised manuscript.

-To analyze the results in clinical strains with mechanisms of resistance involve the overexpression of the efflux pumps

Response: Thanks for your valuable comment.

Bacteria resist the effects of antibiotics mainly through the following molecular mechanisms: modification of the target site, destruction of the antibiotic, antibiotic efflux via efflux transporters and reduced antibiotic influx through decreasing membrane permeability²⁶. Since initial discovery in 1980s, many efflux pumps have been characterized in pathogens such as *Staphylococcus aureus*, *Enterococcus faecium*, *Acinetobacter baumannii*, *Pseudomonas aeruginosa*, and *Klebsiella pneumoniae*²⁷. Overexpression of some efflux pumps can cause clinically relevant levels of antibiotic resistance in Gram-negative pathogens²⁸. The content has added in to the “Introduction” section of the revised manuscript.

-To study the results in clinical strains tige cycline-resistant by overexpression of the efflux pumps.

Response: Thanks for your constructive suggestion. To date, there are several known mechanisms associated with tige cycline-resistant *Klebsiella pneumonia* (TRKP). The most common mechanism is the overproduction of non-specific active resistance-nodulation-cell division (RND) efflux pumps such as AcrAB-TolC¹⁵. Tige cycline MICs of >2 mg/L have been associated with significantly increased ramA levels²⁹. It was also reported that tige cycline resistance in most of the CRKP isolates was associated with increased expression of the efflux pump-encoding *acrB* gene¹⁵. A retrospective study in China displayed that the elevated expression of *acrB* and *ramA* (the regulatory gene of AcrAB-TolC) was found in ~90% of the tige cycline-resistant isolates clinically³⁰. The content has added in to the “Introduction” section of the revised manuscript.

Reference:

1. Xu XL, *et al.* Sialic acid-modified chitosan oligosaccharide-based biphasic calcium phosphate promote synergetic bone formation in rheumatoid arthritis therapy. *J Control Release* **323**, 578-590 (2020).
2. Kang X-Q, *et al.* Tocopherol polyethylene glycol succinate-modified hollow silver nanoparticles for combating bacteria-resistance. **7**, 2520-2532 (2019).
3. Gutbier B, *et al.* Prognostic and Pathogenic Role of Angiopoietin-1 and -2 in Pneumonia. *American journal of respiratory and critical care medicine* **198**, 220-231 (2018).
4. Gebremariam T, *et al.* Preserving Vascular Integrity Protects Mice against Multidrug-Resistant Gram-Negative Bacterial Infection. *Antimicrobial agents and chemotherapy* **64**, (2020).
5. Fenaroli F, *et al.* Enhanced Permeability and Retention-like Extravasation of Nanoparticles from the Vasculature into Tuberculosis Granulomas in Zebrafish and Mouse Models. *ACS nano* **12**, 8646-8661 (2018).
6. Kang X-Q, *et al.* Effective targeted therapy for drug-resistant infection by

- ICAM-1 antibody-conjugated TPGS modified β -Ga₂O₃: Cr³⁺ nanoparticles. **9**, 2739 (2019).
7. Dash R, Bhattacharjya S. Thanatin: An Emerging Host Defense Antimicrobial Peptide with Multiple Modes of Action. *Int J Mol Sci* **22**, (2021).
 8. Ma B, *et al.* The antimicrobial peptide thanatin disrupts the bacterial outer membrane and inactivates the NDM-1 metallo- β -lactamase. *Nat Commun* **10**, 3517 (2019).
 9. Kang XQ, *et al.* Effective targeted therapy for drug-resistant infection by ICAM-1 antibody-conjugated TPGS modified β -Ga(2)O(3):Cr(3+) nanoparticles. *Theranostics* **9**, 2739-2753 (2019).
 10. Vetterli SU, *et al.* Thanatin targets the intermembrane protein complex required for lipopolysaccharide transport in Escherichia coli. **4**, eaau2634 (2018).
 11. Ma B, *et al.* The antimicrobial peptide thanatin disrupts the bacterial outer membrane and inactivates the NDM-1 metallo- β -lactamase. **10**, 1-11 (2019).
 12. Dash R, Bhattacharjya S. Thanatin: An emerging host defense antimicrobial peptide with multiple modes of action. **22**, 1522 (2021).
 13. Sinha S, Zheng L, Mu Y, Ng WJ, Bhattacharjya S. Structure and interactions of a host defense antimicrobial peptide thanatin in lipopolysaccharide micelles reveal mechanism of bacterial cell agglutination. **7**, 1-13 (2017).
 14. Lee M-K, Cha L-N, Lee S-H, Hahm K-S. Role of amino acid residues within the disulfide loop of thanatin, a potent antibiotic peptide. **35**, 291-296 (2002).
 15. Park Y, Choi Q, Kwon GC, Koo SH. Molecular epidemiology and mechanisms of tigecycline resistance in carbapenem-resistant Klebsiella pneumoniae isolates. **34**, e23506 (2020).
 16. Casalone E, *et al.* Characterization of substituted piperazines able to reverse MDR in Escherichia coli strains overexpressing resistance-nodulation-cell division (RND) efflux pumps. *The Journal of antimicrobial chemotherapy* **77**,

413-424 (2022).

17. Magnet S, Courvalin P, Lambert TJAa, chemotherapy. Resistance-nodulation-cell division-type efflux pump involved in aminoglycoside resistance in *Acinetobacter baumannii* strain BM4454. **45**, 3375-3380 (2001).
18. Wang-Kan X, *et al.* Lack of AcrB Efflux Function Confers Loss of Virulence on *Salmonella enterica* Serovar Typhimurium. *mBio* **8**, (2017).
19. Casalone E, *et al.* 1-benzyl-1,4-diazepane reduces the efflux of resistance-nodulation-cell division pumps in *Escherichia coli*. *Future microbiology* **15**, 987-999 (2020).
20. Hayashi K, *et al.* AcrB-AcrA Fusion Proteins That Act as Multidrug Efflux Transporters. *Journal of bacteriology* **198**, 332-342 (2016).
21. Tian Y, Zhang Q, Wen L, Chen JJMS. Combined effect of Polymyxin B and Tigecycline to overcome Heteroresistance in Carbapenem-Resistant *Klebsiella pneumoniae*. **9**, e00152-00121 (2021).
22. Andersson DI, Nicoloff H, Hjort KJNRM. Mechanisms and clinical relevance of bacterial heteroresistance. **17**, 479-496 (2019).
23. Jo J, Ko KSJMs. Tigecycline heteroresistance and resistance mechanism in clinical isolates of *Acinetobacter baumannii*. **9**, e01010-01021 (2021).
24. Hubbard A, *et al.* Piperacillin/tazobactam resistance in a clinical isolate of *Escherichia coli* due to IS26-mediated amplification of blaTEM-1B. **11**, 1-9 (2020).
25. Yang T, Mei H, Wang J, Cai YJFim. Therapeutic Drug Monitoring of Tigecycline in 67 Infected Patients and a Population Pharmacokinetics/Microbiological Evaluation of *A. baumannii* Study. **12**, 1521 (2021).
26. Laws M, Shaaban A, Rahman KMJFMR. Antibiotic resistance breakers: current approaches and future directions. **43**, 490-516 (2019).
27. Lamut A, Peterlin Mašič L, Kikelj D, Tomašič TJMRR. Efflux pump inhibitors of clinically relevant multidrug resistant bacteria. **39**, 2460-2504

(2019).

28. Blair JM, Richmond GE, Piddock LJ. Multidrug efflux pumps in Gram-negative bacteria and their role in antibiotic resistance. *Future microbiology* **9**, 1165-1177 (2014).
29. Pournaras S, Koumaki V, Spanakis N, Gennimata V, Tsakris AJJoaa. Current perspectives on tigecycline resistance in Enterobacteriaceae: susceptibility testing issues and mechanisms of resistance. **48**, 11-18 (2016).
30. Jiang Y, Yang S, Deng S, Lu W, Huang Q, Xia YJJoGAR. Epidemiology and mechanisms of tigecycline-and carbapenem-resistant Enterobacter cloacae in Southwest China: a five-year retrospective study. (2022).

REVIEWER COMMENTS

Reviewer #1 (Remarks to the Author):

Tigecycline (TIG)-resistant *Klebsiella pneumoniae* is difficult to treat. The authors developed calcium phosphate (Cap) nanorods consisting of S-thanatin peptide (Ts)-conjugated tocopheryl polyethylene glycol succinate (TPGS) to encapsulate TIG, and applied the final product, Ts-TPGS/Cap/TIG (TTCT), to overcome therapy resistance. They described preparation and characterization of the drug delivery system, evaluated antibacterial activity in vitro and in vivo, explored mechanism of drug action, and assessed potential toxicity from TTCT. While the antibacterial activity seems convincing, the work lacks details and proper controls in key studies are missing.

1. There is no experiment to properly assign the roles from individual components in the nanorods on drug action. How are nanorods transported to the target organ? Are they internalized by cells in the lung? Which efflux pump is inhibited (AcrAB, OqxAB, or ramA)? By free TPGS, or free Ts-TPGS, or Ts-TPGS in nanorods?
2. Fig. 1. There are red circles and squares, but no text to explain them in the figure legend.
3. Fig 2. Over 50% drug molecules got released within the first hour in the in vitro drug release test (Fig. 2G). The result indicates that the majority of drug molecules will be released before the nanorods get accumulated in the lung. Do nanorods disassemble in circulation? This seems to be a design defect. In addition, there is no scale bar in Fig. 2D.
4. Fig. 4A-D. A non-targeting peptide-conjugated TPGS/Cap should serve as a proper control.
5. Fig. 4E. EB accumulation served as an indicator for drug accumulation. Is EB subjected to the same efflux pump(s) as TIG?
6. Fig. 7. Very fuzzy PET-CT images.

Reviewer #2 (Remarks to the Author):

The work titled "TPGS-based and S-thanatin functionalized nanorods for overcoming drug resistance in *Klebsiella pneumoniae*" showed as the use combined of Ts-TPGS/Cap nanocarrier produce targeted accumulation in bacteria of tigecycline and exhibited antibacterial capacity both in vitro and in vivo. The obtained TIG-loaded Ts-TPGS/Cap nanorods (TTCT) could enhance tigecycline accumulation in bacteria

via an inhibitory effect on efflux pumps and the targeting capacity to bacteria, thereby showing excellent antibacterial effects and overcoming the drug resistance of TRKP.

The work is very interesting but I have the following considerations to take in account the authors:

Major considerations

The study shows important results in relation a in vitro e in vivo experiments but must carry out more experiments about microbiology field to confirm the conclusion of the accumulation of tigecycline in the bacteria by inhibition of the efflux pumps through Ts-TPGS/Cap nanocarrier.

The authors must analyze the protein expression of the efflux pumps and the putative mechanisms of activity synergy by checkerboard assays (to calculate fractional inhibitory concentration index [FICI]) and Optical densities growth curves.

Moreover, the inhibitors of the efflux pumps could produce heteroresistance or resistance mechanisms associated to mutations in regulatory genes. It must be analyzed due to Ts-TPGS/Cap nanocarrier could not to success in the treatment of patients.

Minor considerations

-To improve the quality of the Abstract Figure

-To analyze the results in clinical strains with mechanisms of resistance involve the overexpression of the efflux pumps

-To study the results in clinical strains tigecycline-resistant by overexpression of the efflux pumps.

REVIEWERS' COMMENTS

Reviewer #3 (Remarks to the Author):

The authors have followed all comments from reviewers that let it improving the manuscript with news results.

Response: Thank you very much. We are so delighted to receive your positive comment.

Reviewer #4 (Remarks to the Author):

The reviewer's concerns are properly addressed and the quality of the manuscript is improved substantially after revision. It is recommended for publication in Nature Communications after the following issues can be addressed:

1. To improve the reproducibility of the experiments, the authors should provide the source and No. of the strains, and the bacterial dose for each challenge experiment.

Response: Thanks for you comment. The KPN and TRKP strains we used were clinical isolates and kindly provided by State Key Laboratory for Diagnosis and Treatment of Infectious Diseases, The First Affiliated Hospital, College of Medicine, Zhejiang University (Hangzhou, China). The source of the strains has been added to the “Materials” section of the revised manuscript. To ensure consistency of the experiments, we performed the experiments with one strain throughout the research. The bacterial dose for each challenge experiment has been clarified in the method section.

2. In Figure 4A, the particle size of the nanorods is obviously smaller than 20 μm , which is inconsistent with the results in Figure 2B.

Response: Thanks so much for your comment. The scale bar of Figure 4A was 20 μm , and the particle size of the nanorods was maintained at ~ 25 nm (Table 1 and Figure 2B). The displayed data had different orders of magnitude. Therefore, the results were consistent, and the particle size of the nanorods (~ 25 nm) was indeed obviously smaller than 20 μm .

3. Please add statistical analysis to Figures 6A, 6B and 6I.

Response: Thanks so much for your comment. Log-rank (Mantel-Cox) test was performed in Figure 6A and B, and the results of statistical analysis have been added in the revised manuscript. One-way analysis of variance (ANOVA) with post hoc Tukey tests were performed for Figure 6I, and the corresponding contents were supplied as supplementary Fig. 3 in the Supplementary information.

4. The authors should discuss whether the TTCT system has advantages against highly virulent *Klebsiella pneumoniae*.

Response: Thanks so much for your comment. As reported, similar to classical *K. pneumoniae* (cKp), hypervirulent *K. pneumoniae* (hvKp) strains are becoming increasingly resistant to antimicrobials *via* acquisition of mobile elements carrying resistance determinants. When extensively drug-resistant cKp strains acquire hvKp-specific virulence determinants, new hvKp strains emerge and result in nosocomial infection¹.

Considering the potential severity of hvKp infection and its propensity for metastatic spread, tigecycline plays important roles in the empirical treatment of carbapenem-resistant hvKp strains. However, tigecycline resistance has been reported for an cKp strain that evolved into an hvKP strain

via acquisition of a portion of an hvKp virulence plasmid. Overexpression of the efflux pump gene *acrR* and its regulatory gene *ramA* has been demonstrated to be associated with tigecycline resistance². Based on our data, we speculate that TTCT system still has advantages against carbapenem-resistant hvKp strains.

For tigecycline-sensitive hvKp, TTCT exerts more than one distinct mechanism and the synergistic/addition activity between Ts and TIG might reduce the likelihood of bacteria developing resistance. Tigecycline is a time-dependent antimicrobial agent with long post-antibiotic effect (PAE), and its pharmacokinetics/pharmacodynamics (PK/PD) is assessed by AUC/MIC. For tigecycline-resistant hvKp, the increase of MIC and the limiting tigecycline dosage were the main reasons for the poor clinical efficacy³. The MIC reduction endowed by TTCT would be beneficial for maximizing the efficacy and minimizing the drug-related adverse events of tigecycline.

The corresponding content has been added to the revised manuscript (Line 466-490)

5. In Figure 6I, the author should explain that the bacterial load in BALF was detected instead of in lung homogenate.

Response: Thanks so much for your comment. Bronchoalveolar lavage fluid (BALF) has been widely utilized for pathogen identification in pneumonia patients clinically^{4,5,6} and experimental animals⁷. Therefore, BALF was utilized for bacterial colony analysis in our research. This has been clarified in the method section (Line 714-715) and the legend of Figure 6 (Line 1071) in the revised manuscript

6. There are some clerical errors in the paper. For example, the name of the group in Figure 3D is wrong. The "in vivo" and "in vitro" should be italic.

Response: Thanks so much for your comment. We feel so sorry for the mistakes. The name of the group in Figure 3D has been corrected in the revised manuscript. The "in vivo" and "in vitro" have been changed to italics in the revised manuscript.

Reference:

1. Russo TA, Marr CM. Hypervirulent *Klebsiella pneumoniae*. *Clinical microbiology reviews* **32**, (2019).
2. Huang YH, *et al.* Emergence of an XDR and carbapenemase-producing hypervirulent *Klebsiella pneumoniae* strain in Taiwan. *The Journal of antimicrobial chemotherapy* **73**, 2039-2046 (2018).
3. Yang T, Mei H, Wang J, Cai Y. Therapeutic Drug Monitoring of Tigecycline in 67 Infected Patients and a Population Pharmacokinetics/Microbiological Evaluation of *A. baumannii* Study. *Frontiers in microbiology* **12**, 678165 (2021).
4. Zhou H, *et al.* Clinical Impact of Metagenomic Next-Generation Sequencing of Bronchoalveolar Lavage in the Diagnosis and Management of Pneumonia: A Multicenter Prospective Observational Study. *The Journal of molecular diagnostics : JMD* **23**, 1259-1268 (2021).
5. Chen Y, *et al.* Application of Metagenomic Next-Generation Sequencing in the Diagnosis of Pulmonary Infectious Pathogens From Bronchoalveolar Lavage Samples. *Frontiers in cellular and infection microbiology* **11**, 541092 (2021).
6. Azoulay E, *et al.* Diagnosis of severe respiratory infections in immunocompromised patients.

Intensive care medicine **46**, 298-314 (2020).

7. Curran M, Boothe DM, Hathcock TL, Lee-Fowler T. Analysis of the effects of storage temperature and contamination on aerobic bacterial culture results of bronchoalveolar lavage fluid. *Journal of veterinary internal medicine* **34**, 160-165 (2020).